# Locally Subspace-Informed Neural Operators for Efficient Multiscale PDE Solving

**Alexander Rudikov**
AXXX,
Institute of Numerical Mathematics
ay.rudikov@gmail.com

**Vladimir Fanaskov**
AXXX, Applied AI Institute

**Sergei Stepanov**
North-Eastern Federal University

**Buzheng Shan**
Department of Mathematics,
Texas A&M University

**Ekaterina Muravleva**
Institute of Numerical Mathematics

**Yalchin Efendiev**
Department of Mathematics,
Texas A&M University

**Ivan Oseledets**
AXXX,
Institute of Numerical Mathematics

## Abstract

We propose GMsFEM-NO, a novel hybrid framework that combines the robustness of the Generalized Multiscale Finite Element Method (GMsFEM) with the computational speed of neural operators (NOs) to create an efficient method for solving heterogeneous partial differential equations (PDEs). GMsFEM builds localized spectral basis functions on coarse grids, allowing it to capture important multiscale features and solve PDEs accurately with less computational effort. However, computing these basis functions is costly. While NOs offer a fast alternative by learning the solution operator directly from data, they can lack robustness. Our approach trains a NO to instantly predict the GMsFEM basis by using a novel subspace-informed loss that learns the entire relevant subspace, not just individual functions. This strategy significantly accelerates the costly offline stage of GMsFEM while retaining its foundation in rigorous numerical analysis, resulting in a solution that is both fast and reliable. On standard multiscale benchmarks—including a linear elliptic diffusion problem and the nonlinear, steady-state Richards equation—our GMsFEM-NO method achieves a reduction in solution error compared to standalone NOs and other hybrid methods. The framework demonstrates effective performance for both 2D and 3D problems. A key advantage is its discretization flexibility: the NO can be trained on a small computational grid and evaluated on a larger one with minimal loss of accuracy, ensuring easy scalability. Furthermore, the resulting solver remains independent of forcing terms, preserving the generalization capabilities of the original GMsFEM approach. Our results prove that combining NO with GMsFEM creates a powerful new type of solver that is both fast and accurate.

## 1 Introduction

Many practical multiscale problems involve highly heterogeneous properties with high-contrast variations across multiple scales, posing significant challenges for the numerical solution of partial differential equations (PDEs). A well-established approach for such problems is the Generalized Multiscale Finite Element Method (GMsFEM) Efendiev et al. (2011; 2013); Chung et al. (2016),

which constructs localized spectral basis functions on coarse grids. By solving local eigenproblems, GMsFEM captures fine-scale information, enabling accurate coarse-scale solutions. However, this accuracy comes at a high computational cost due to the expense of solving these local eigenproblems.

Recently, data-driven solvers, particularly neural operators (NOs) like Fourier Neural Operators (FNOs) Li et al. (2020); Kovachki et al. (2023); Fanaskov & Oseledets (2023); Tran et al. (2021) and DeepONets Lu et al. (2021); Wang et al. (2021), have emerged as a powerful alternative for accelerating PDE simulations Azizzadenesheli et al. (2024); Karniadakis et al. (2021). While effective for problems with smooth coefficients, standard NOs often struggle to efficiently capture the localized features of high-contrast heterogeneities, typically requiring extensive data and large network architectures.

In this work, we introduce GMsFEM-NO, a hybrid framework that combines the robustness of GMsFEM with the speed of neural operators. Our key innovation is a subspace-informed NO that learns to map a heterogeneous coefficient field directly to the low-dimensional subspace spanned by the GMsFEM basis functions. Instead of learning individual basis functions—which can be sensitive to small perturbations—we design a novel subspace-aware loss function that enforces physical consistency at the subspace level. This approach offers several advantages: it is more data-efficient than learning the full PDE solution, as the basis functions are smoother and of lower dimension; and it is more robust than a pure NO, as the final solution is obtained through a GMsFEM, ensuring legitimacy even with imperfect basis predictions.

Our approach is distinct from existing hybrid methods Bhattacharya et al. (2024); Vasilyeva et al. (2020); Wang et al. (2020); Liu et al. (2023); Kröpfl et al. (2022; 2025) that combine machine learning with numerical homogenization/upscaling/macroscopic-modeling. Those methods typically assume a known macroscopic equation form and learn effective coefficients, which is infeasible for problems without scale separation and with high contrast. In contrast, GMsFEM-NO learns the macroscopic solution space itself, in the form of multiscale basis functions, making it suitable for these more challenging settings. A related approach Spiridonov et al. (2025) used a fully connected neural network to predict an additional basis function for the steady-state Richards equation Richards (1931); Farthing & Ogden (2017), supplementing an existing set of precomputed basis functions. While this approach enhanced prediction accuracy, it failed to deliver computational efficiency gains because traditional methods still generated most basis functions. Furthermore, the simplicity of the fully connected architecture limited its ability to account for spatial variations, potentially compromising prediction accuracy for high-contrast data. Another category of related work aims to reduce the computational cost of PDE solving via reduced-order modeling (POD Volkwein (2013), DeepPOD Franco et al., and PCANet Bhattacharya et al. (2021)). DeepPOD and PCANet also leverage neural networks to learn compact solution representations, providing a relevant baseline for comparing the efficiency of our method.

We validate GMsFEM-NO on two challenging benchmarks with high-contrast coefficients: a linear elliptic diffusion problem and the nonlinear steady-state Richards equation. Results showed that our approach is better than NO in terms of solution accuracy and requires less training data to achieve similar accuracy. Additionally, it reduces basis-construction time by more than 60 times compared to traditional GMsFEM.

Our main contributions are:

1. We introduce a novel hybrid approach (GMsFEM-NO) that combines the strengths of NOs with GMsFEM (see Fig. 1).

2. A new subspace-informed loss function for learning stable and generalizable solution subspaces.

3. The approach is evaluated on high-contrast PDEs and shown to deliver the same results as GMsFEM at a fraction of the computational cost.

4. Demonstration of resolution invariance of GMsFEM-NO: effective training on low-resolution data for application to high-resolution problems.

5. Superior in-distribution and out-of-distribution performance compared to standard NOs, without requiring domain adaptation.

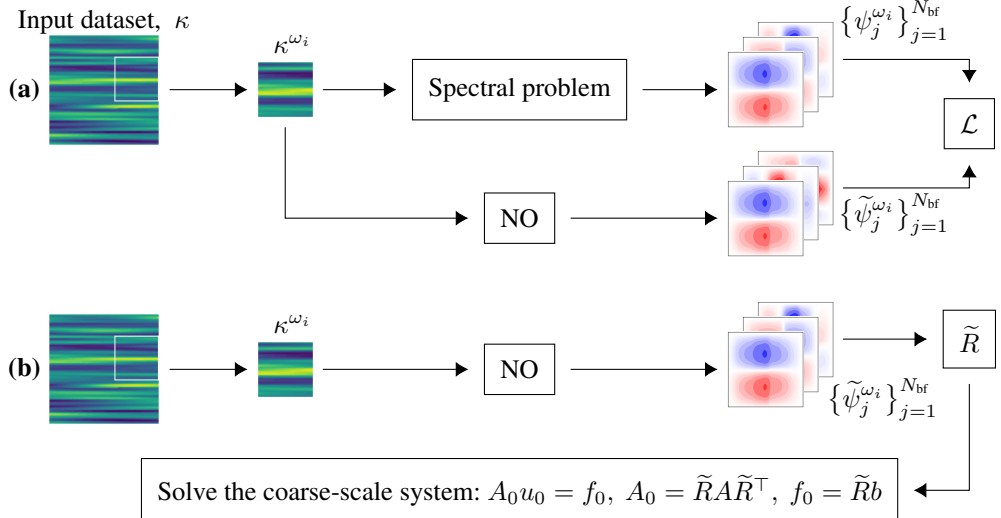

Figure 1: Illustration of training **(a)** and inference **(b)** stages of the proposed GMsFEM-NO method. NO is trained on heterogeneous fields $\kappa^{\omega_i}$ that defined on subdomain $\omega_i$ to predict subspace of basis functions $\{\psi_j^{\omega_i}\}_{j=1}^{N_{\text{bf}}}$, where $N_{\text{bf}}$ is the number of basis functions. During training the subspace-informed loss $\mathcal{L}$ is applied to align predicted subspace $\{\widetilde{\psi}_j^{\omega_i}\}_{j=1}^{N_{\text{bf}}}$ with $\{\psi_j^{\omega_i}\}_{j=1}^{N_{\text{bf}}}$. During inference stage **(b)**, the predicted subspace forms the matrix $\widetilde{R}$ that projects matrix $A$ and vectors to the coarse space.

## 2 LOCALLY SUBSPACE-INFORMED NEURAL OPERATORS

### 2.1 DIFFUSION EQUATION

We consider the diffusion equation with heterogeneous coefficient

$$-\nabla \cdot \big(\kappa(x)\nabla u(x)\big) = f(x), \quad x \in \Omega \equiv (0,\ 1)^D,\ u(x)\big|_{x \in \partial\Omega} = 0, \tag{1}$$

where $\partial\Omega$ is a boundary of the unit hypercube $\Omega$, and $\kappa(x)$ is a heterogeneous field with high contrast. In particular, we assume that $\kappa(x) \geqslant \varepsilon > 0$, while $\kappa(x)$ can have very large variations. For example, in this work we use $\kappa(x)$ with values in range $[1,\ 9600]$.

### 2.2 STEADY-STATE RICHARDS' EQUATION

The steady-state version of Richards' equation, which describes water movement in unsaturated porous media, takes the following form:

$$-\nabla \cdot \big(\kappa\big(x, u(x)\big)\nabla u(x)\big) = f(x), \quad x \in \Omega \equiv (0,\ 1)^D,\ u(x)\big|_{x \in \partial\Omega} = 0, \tag{2}$$

where $\kappa\big(x, u(x)\big)$ is unsaturated hydraulic conductivity, $u(x)$ is the water pressure and $f(x)$ is a source or sink term.

We consider the Haverkamp model Haverkamp et al. (1977) to define $\kappa\big(x, u(x)\big)$:

$$\kappa\big(x, u(x)\big) = K_s(x)\, K_r(u(x)) = \kappa(x)\, \frac{1}{1 + |u|},$$

where $\kappa(x)$ is a heterogeneous field with high contrast that denotes the permeability of soils, $K_r(u)$ represents the relative hydraulic conductivity, $K_s(x)$ stands for the saturated hydraulic conductivity. All the multiscale heterogeneity is incorporated in $\kappa(x)$ without regard to $u$, and $\dfrac{1}{1 + |u|}$ includes all the non-linearity.

## 2.3 GENERALIZED MULTISCALE FINITE ELEMENT METHOD

### 2.3.1 MULTISCALE SPACE APPROXIMATION

Multiscale methods Efendiev & Hou (2009) form a broad class of numerical techniques. They are based on constructing multiscale basis functions in local domains to capture fine-scale behavior.

Let $\mathcal{T}_H$ be a coarse mesh of the domain $\Omega \subset \mathbb{R}^D$ (with $D = 2$ or 3), such that $\mathcal{T}_H = \bigcup_{i=1}^{N_c} K_i$, where each $K_i$ is a coarse cell and $N_c$ is the number of coarse cells. Let $\mathcal{T}_h$ be a fine grid obtained by a refinement of $\mathcal{T}_H$, with $h \ll H$. We denote by $\{x_i\}_{i=1}^{N_v}$ the nodes of the coarse mesh $\mathcal{T}_H$, where $N_v$ is the number of nodes of the coarse mesh. Let $\omega_i$ be the subdomain defined as the collection of coarse cells containing the coarse grid node $x_i$ (see Fig. 2 in Appendix A):

$$\omega_i = \bigcup_j \left\{ K_j \in \mathcal{T}_H : x_i \in \overline{K}_j \right\}.$$

To ensure accurate approximations on the coarse mesh $\mathcal{T}_H$, we construct spectral multiscale basis functions following the GMsFEM. GMsFEM contains two stages:

**Offline stage:**

1. **Coarse and Local Domain Definition:** Define the coarse grid $\mathcal{T}_H$ and generate the associated local domains $\omega_i$ for $i = 1, \dots, N_v$.

2. **Local Spectral Problem Solving:** In each local domain $\omega_i$, solve a local spectral problem to obtain a set of eigenvectors $\left\{ \phi_j^{\omega_i} \right\}_{j=1}^N$, where $N$ is the number of coarse eigenvectors.

3. **Multiscale Basis Function Construction:** Select the first $N_{\text{bf}}$ eigenvectors from each $\omega_i$ and multiply them by a partition of unity function $\chi_i$ Babuska & Lipton (2011); Babuška et al. (2008); Strouboulis et al. (2000) to create the final multiscale basis functions $\left\{ \psi_j^{\omega_i} \right\}_{j=1}^{N_{\text{bf}}}$, where $N_{\text{bf}} \leqslant N$.

4. **Global System Assembly:** Map the local degrees of freedom to global and form a restriction matrix $R$.

**Online stage:**

1. **Projection:** Use $R$ to project the fine-scale system onto the coarse space.

2. **Solution:** Compute the solution within the coarse multiscale space.

3. **Reconstruction:** Obtain the fine-scale approximation by applying the prolongation operator $R^\top$ to the coarse-scale solution.

### 2.3.2 SPECTRAL PROBLEM

We denote by $V^h(\Omega)$ the usual finite element discretization of piecewise linear continuous functions with respect to the fine grid $\mathcal{T}_h$. For each local domain $\omega_i$, we define the **Neumann** matrix $A_h^{\omega_i}$ by

$$v_h^\top A_h^{\omega_i} w_h = \int_{\omega_i} \kappa(x) \nabla v_h \cdot \nabla w_h \, dx, \quad \forall v_h, w_h \in V^h(\omega_i)$$

and the **Mass** matrix $S_h^{\omega_i}$ by

$$v_h^\top S_h^{\omega_i} w_h = \int_{\omega_i} \kappa(x) v_h w_h \, dx, \quad \forall v_h, w_h \in V^h(\omega_i).$$

We consider the finite dimensional symmetric eigenvalue problem

$$A_h^{\omega_i} \phi = \lambda S_h^{\omega_i} \phi$$

and denote its eigenvalues and eigenvectors by $\left\{ \lambda_j^{\omega_i} \right\}_{j=1}^N$ and $\left\{ \phi_j^{\omega_i} \right\}_{j=1}^N$, respectively. Note that $\lambda_1^{\omega_i} = 0$ corresponds to the constant eigenvector $\phi_1^{\omega_i} = \text{const}$. We order eigenvalues as

$$\lambda_1^{\omega_i} \leqslant \lambda_2^{\omega_i} \leqslant \dots \leqslant \lambda_j^{\omega_i} \leqslant \dots .$$

The eigenvectors $\left\{ \phi_j^{\omega_i} \right\}_{j=1}^N$ form an $S_h^{\omega_i}$-orthonormal basis of $V^h(\omega_i)$.

### 2.3.3 SOLVING OF THE COARSE-SCALE SYSTEM

For each local domain $\omega_i$, we select eigenvectors corresponding to the $N_{\text{bf}} \leqslant N$ smallest eigenvalues and define a multiscale subspace

$$\text{span}\big\{\psi_j^{\omega_i} = \chi_i \phi_j^{\omega_i} \,\big|\, j = 1, \ldots, N_{\text{bf}}, \ i = 1, \ldots, N_v\big\} \tag{3}$$

and define the restriction matrix $R^\top = \left[\psi_1^{\omega_1}, \ldots, \psi_{N_{\text{bf}}}^{\omega_1}, \ldots, \psi_1^{\omega_{N_v}}, \ldots, \psi_{N_{\text{bf}}}^{\omega_{N_v}}\right]$. Coarse-grid solution is the finite element projection of the fine-scale solution into the space (3). More precisely, multiscale solution $u_0$ is given by

$$A_0 u_0 = f_0,$$

where $A_0 = RAR^\top$ is the projected system matrix, $f_0 = R^\top b$ is projected right-hand side. The reconstructed fine-scale solution is $u = R^\top u_0$.

### 2.4 NEURAL OPERATOR

Here we consider one type of NO that employs Fourier modes, but there are no restrictions on using other types of NOs. Fourier neural operators (FNOs) are a class of NOs motivated by Fourier spectral methods. Originally, Li et al. (2020) formulate each operator layer as

$$\mathcal{L}^\ell\big(z^{(\ell)}\big) = \sigma\Big[W^{(\ell)} z^{(\ell)} + b^{(\ell)} + \mathcal{K}^{(\ell)}\big(z^{(\ell)}\big)\Big], \tag{4}$$

where $W^{(\ell)} z^{(\ell)} + b^{(\ell)}$ is an affine point-wise map,

$$\mathcal{K}^{(\ell)}\big(z^{(\ell)}\big) = \text{IFFT}\Big(\mathcal{R}^{(\ell)} \cdot \text{FFT}(z)\Big)$$

is a kernel integral operator. The Fourier domain weight matrices $\big\{\mathcal{R}^{(\ell)}\big\}_{\ell=1}^L$ require $O(LH^2 M^D)$ parameters, where $H$ is the hidden size, $M$ is the number of the top Fourier modes that are kept, and $D$ is the dimension of the problem.

In Factorised FNO (F-FNO) Tran et al. (2021), the operator layer in (4) is changed

$$\mathcal{L}^\ell\big(z^{(\ell)}\big) = z^{(\ell)} + \sigma\Big[W_2^{(\ell)} \sigma\Big(W_1^{(\ell)} \mathcal{K}^{(\ell)}\big(z^{(\ell)}\big) + b_1^{(\ell)}\Big) + b_2^{(\ell)}\Big],$$

where $\mathcal{K}^{(\ell)}\big(z^{(\ell)}\big) = \sum_{d \in D} \Big[\text{IFFT}\Big(\mathcal{R}_d^{(\ell)} \cdot \text{FFT}_d\big(z^{(\ell)}\big)\Big)\Big]$. In this case, the number of parameters is $O(LH^2 MD)$. Therefore, the FFNO reduces model complexity and scales efficiently to deeper networks.

### 2.5 PROPOSED METHOD

#### 2.5.1 GMsFEM-NO ALGORITHM

We propose an efficient hybrid method for generating basis functions in the GMsFEM using NOs, significantly accelerating the offline stage.

Local domains vary in shape and orientation (see Appendix A), where orientation refers to the relative placement of the coarse node $x_i$ shared by all cells in the local domain. We address this variability by categorizing the local domains based on their geometry: into **full**, **half**, and **corner** types in 2D, and into **full**, **half**, **quarter**, and **corner** types in 3D (see Appendix A). Before training, we normalize the orientation of each local domain by rotating both the input data and the target basis functions, ensuring a standardized coarse node $x_i$ position within each group. This preprocessing step guarantees consistency in the input structure for the NO.

We train separate NOs, each specialized for one domain group (see Appendix B). Each NO predicts $N_{\text{bf}}$ basis functions for local domains within its assigned category. This group-specific approach improves prediction accuracy by accounting for geometric variations across local domain types.

For test data, we first decompose the computational domain into local domains. The corresponding NO then generates the required basis functions. The predicted basis functions are extended to the domain $\Omega$ (with zeros padded outside their respective local domains) and vectorized to construct the

restriction matrix $R$. Finally, the online stage of GMsFEM is executed to compute the multiscale solution.

This approach substantially reduces offline computational costs while maintaining the accuracy and flexibility of GMsFEM, making it particularly suitable for problems with heterogeneous or highly varying coefficients.

### 2.5.2 SUBSPACE-INFORMED LOSS FUNCTIONS

The selection of an appropriate loss function is critical when training NOs. We propose a **Subspace Alignment Loss (SAL)** that directly optimizes the geometric consistency of the learned subspaces. Let $R^i = \left[\psi_1^i, \ldots, \psi_{N_{\text{bf}}}^i\right]^\top$ represent the target subspace basis and $\widetilde{R}^i$ denote the predicted subspace. The SAL measures alignment between subspaces using their orthonormalized bases $Q_{R^i}$ and $Q_{\widetilde{R}^i}$:

$$\mathcal{L}_{\text{SAL}} = \mathbb{E}_i\left[N_{\text{bf}} - \left\|Q_{R^i}^\top Q_{\widetilde{R}^i}\right\|_F^2\right], \tag{5}$$

where the Frobenius norm term $\|Q_{R^i}^\top Q_{\widetilde{R}^i}\|_F^2$ quantifies the subspace overlap, achieving its maximum value $N_{\text{bf}}$ when subspaces are perfectly aligned. Appendix C proves SAL's theoretical foundation and provides error bounds connecting subspace alignment to solution accuracy.

While SAL ensures subspace coherence, it may overlook finer discrepancies in how functions are projected onto the subspaces. To enforce consistency in projection behavior, we introduce a **Projection Regularization** term. This term evaluates the discrepancy between projections of a randomized test vector $v^i$ onto the target and predicted subspaces, governed by their projection matrices $P_{R^i}$ and $P_{\widetilde{R}^i}$:

$$\mathcal{L}_{\text{SAL-PR}} = \mathcal{L}_{\text{SAL}} + \lambda \cdot \mathbb{E}_{i,c}\left\|\left(P_{R^i} - P_{\widetilde{R}^i}\right)v^i\right\|_2^2, \quad c \sim \mathcal{N}(0, I), \tag{6}$$

where $v^i = \sum_{k=1}^{N_{\text{bf}}} c_k\psi_k^i$, $P_{R^i} = Q_{R^i}Q_{R^i}^\top$, $P_{\widetilde{R}^i} = Q_{\widetilde{R}^i}Q_{\widetilde{R}^i}^\top$, and $\lambda$ is a hyperparameter.

We compare proposed loss functions (5), (6) with conventional one which is $L_2$ loss. Since basis functions are defined only up to their sign (see Appendix E), the conventional $L_2$ loss is adapted to account for this invariance, resulting in the Relative Basis Function $L_2$ Loss (RBFL$_2$):

$$\mathcal{L}_{\text{RBFL}_2} = \mathbb{E}_{i,j}\left[\min\left(\frac{\|\psi_j^i - \tilde{\psi}_j^i\|_2^2}{\|\psi_j^i\|_2^2}, \frac{\|\psi_j^i + \tilde{\psi}_j^i\|_2^2}{\|\psi_j^i\|_2^2}\right)\right], \tag{7}$$

where $\psi_j^i$ and $\tilde{\psi}_j^i$ denote the $j$-th target and predicted basis functions for the $i$-th local domain $\omega_i$. The minimization over $\pm\tilde{\psi}_j^i$ ensures invariance to sign permutations.

## 3 RESULTS

We use datasets of 2D coefficients at resolutions of $100^2$ and $250^2$, and 3D coefficients at $50^3$ and $100^3$ (see example in Appendix F). The domain is partitioned into $N_v$ subdomains corresponding to coarse grids (e.g., $N_v = 36$ for $5 \times 5$, 121 for $10 \times 10$, 216 for $5 \times 5 \times 5$, 729 for $8 \times 8 \times 8$). For all grid sizes except $100 \times 100 \times 100$, we used 1000 samples (800 train, 200 test). For the $100 \times 100 \times 100$ grid, we used 150 train and 50 test samples due to computational constraints. The different local domain types occur with varying frequencies within a single sample (see Appendix B). For training NOs, we utilize the first 8 basis functions ($N_{\text{bf}}$) per subdomain as training targets.

To evaluate method robustness, we consider two right-hand side configurations:

- Uniform unit forcing term
$$f(x) = 1. \tag{8}$$

- Spatially variable forcing (see Appendix F) defined by
$$f(x) \sim \gamma \cdot \mathcal{N}\left(\alpha \cdot (I - \Delta)^{-\beta}\right). \tag{9}$$

To measure quality of the obtained solutions on fine grid, we use the following metrics:

$$L_2 = \mathbb{E}_n \left[ \sqrt{\frac{\int_\Omega |u_h^n - \tilde{u}_h^n|^2 dx}{\int_\Omega |u_h^n|^2 dx}} \right], \quad H_1 = \mathbb{E}_n \left[ \sqrt{\frac{\int_\Omega |\nabla u_h^n - \nabla \tilde{u}_h^n|^2 dx}{\int_\Omega |\nabla u_h^n|^2 dx}} \right].$$

All experiments were performed on a single Nvidia Tesla H100 80Gb HBM3. The comparison of our approach with baseline methods is presented in Appendix G.

## 3.1 $\mathcal{L}_{\mathrm{RBFL_2}}$ vs. $\mathcal{L}_{\mathrm{SAL}}$, $\mathcal{L}_{\mathrm{SAL-PR}}$

To determine the optimal training configuration for the NOs predicting basis function subspaces, we performed a grid search over architectural parameters and training hyperparameters. Full specifications (except loss function type) are in Appendix H. Loss function results for $N_{\mathrm{bf}} = 8$ are shown in Table 1. For $N_{\mathrm{bf}} = 4$, results are in Appendix J.

As shown in Table 1, $\mathcal{L}_{\mathrm{RBFL_2}}$ underperforms compared to our proposed subspace alignment losses ($\mathcal{L}_{\mathrm{SAL}}$, $\mathcal{L}_{\mathrm{SAL-PR}}$). For the Richards equation with simple right-hand side (8) and $N_{\mathrm{bf}} = 8$, our proposed loss improves the relative $L_2$ metric by a factor of 1.8. Notably, the projection regularization term in $\mathcal{L}_{\mathrm{SAL-PR}}$ yielded nearly identical results to $\mathcal{L}_{\mathrm{SAL}}$. While projection regularization had a minimal impact on smaller grids—likely because the subspace alignment term alone suffices—its effect became significant for larger problems. For the $250^2$ grid using Richards' equation with right-hand side (8), it reduced the $L_2$ error from $1.82\%$ to $1.72\%$ (see Table 3).

Table 1: Performance comparison of loss functions for NO training ($100 \times 100$ grid, $N_v = 36$).

| $N_{\mathrm{bf}}$ | Dataset | $\mathcal{L}_{\mathrm{RBFL_2}}$ | | $\mathcal{L}_{\mathrm{SAL}}$ | | $\mathcal{L}_{\mathrm{SAL-PR}}$ | |
|---|---|---|---|---|---|---|---|
| | | $L_2$ | $H_1$ | $L_2$ | $H_1$ | $L_2$ | $H_1$ |
| | Diffusion, 8 | 1.75% | 14.83% | **1.06**% | 11.57% | **1.06**% | 11.65% |
| 8 | Diffusion, 9 | 3.53% | 21.77% | 2.82% | 19.07% | **2.81**% | 19.03% |
| | Richards, 8 | 3.46% | 15.04% | 1.88% | 11.10% | **1.87**% | 11.25% |
| | Richards, 9 | 3.77% | 22.38% | **2.99**% | 19.61% | **2.99**% | 19.60% |

## 3.2 GMsFEM vs. GMsFEM-NO

In this section, we compare the performance of the original GMsFEM and our proposed GMsFEM-NO methods in terms of solution accuracy (quantified by $L_2$ and $H_1$ metrics) and computational efficiency for basis functions generation. For large-scale 3D simulations ($100 \times 100 \times 100$ grid), we employed the GMsFEM-NO method with the SAL loss. Although the SAL-PR loss offers benefits for smaller problems (see Appendix D), its computational cost becomes prohibitive at this scale, making the standard SAL loss the practical choice.

As shown in Tables 2, 3, 4 and 5, GMsFEM-NO achieves nearly identical $L_2$ and $H_1$ errors to GMsFEM across all datasets and grid sizes. While GMsFEM-NO shows slightly better results for some configurations, this is likely due to statistical variation. The same behaviour is observed for the time-dependent equation and the diffusion equation with mixed boundary conditions (see Appendix L and M); these experiments were conducted solely on the $250 \times 250$ grid.

Table 6 compares the time required to generate 8 basis functions using the GMsFEM offline stage and GMsFEM-NO for different grid sizes and $N_v$ values. GMsFEM-NO employs several NOs, one for each local domain type. The proposed method achieves more than $60\times$ speedup, demonstrating its computational superiority. Basis calculation speedup grows with grid size and dimensionality.

While GMsFEM-NO reduces the cost of the traditional GMsFEM offline stage over many subsequent simulations, it does not eliminate cost of train data generation and training. The breakeven point, defined as the number of inference samples required to offset these initial costs, is derived in the Appendix N.

Table 2: Performance comparison of GMsFEM and GMsFEM-NO for 2D ($100 \times 100$, $N_v = 36$).

| $N_{bf}$ | Dataset | GMsFEM | | GMsFEM-NO | |
| | | $L_2$ | $H_1$ | $L_2$ | $H_1$ |
|---|---|---|---|---|---|
| | Diffusion, 8 | 1.15% | 11.68% | **1.06**% | 11.57% |
| 8 | Diffusion, 9 | 2.82% | 19.07% | **2.81**% | 19.03% |
| | Richards, 8 | 2.03% | 11.68% | **1.87**% | 11.25% |
| | Richards, 9 | 3.09% | 20.20% | **2.99**% | 19.60% |

Table 3: Performance comparison of GMsFEM and GMsFEM-NO for 2D ($250 \times 250$, $N_v = 121$).

| $N_{bf}$ | Dataset | GMsFEM | | GMsFEM-NO, $\mathcal{L}_{SAL}$ | | GMsFEM-NO, $\mathcal{L}_{SAL\text{-}PR}$ | |
| | | $L_2$ | $H_1$ | $L_2$ | $H_1$ | $L_2$ | $H_1$ |
|---|---|---|---|---|---|---|---|
| | Diffusion, 8 | 1.12% | 14.01% | 1.13% | 13.92% | **1.05**% | 14.07% |
| 8 | Diffusion, 9 | **1.60**% | 21.49% | 1.62% | 22.50% | **1.60**% | 22.03% |
| | Richards, 8 | 1.79% | 14.57% | 1.82% | 14.27% | **1.72**% | 14.53% |
| | Richards, 9 | 1.62% | 21.76% | 1.62% | 22.62% | **1.60**% | 22.13% |

Table 4: Performance comparison of GMsFEM and GMsFEM-NO for 3D ($50 \times 50 \times 50$, $N_v = 216$).

| $N_{bf}$ | Dataset | GMsFEM | | GMsFEM-NO, $\mathcal{L}_{SAL}$ | | GMsFEM-NO, $\mathcal{L}_{SAL\text{-}PR}$ | |
| | | $L_2$ | $H_1$ | $L_2$ | $H_1$ | $L_2$ | $H_1$ |
|---|---|---|---|---|---|---|---|
| | Diffusion, 8 | **3.07**% | 20.72% | 3.10% | 20.39% | 3.10% | 20.39% |
| 8 | Diffusion, 9 | 5.08% | 25.26% | 5.01% | 24.92% | **5.00**% | 24.86% |
| | Richards, 8 | **4.04**% | 15.43% | 4.12% | 15.37% | 4.14% | 15.39% |
| | Richards, 9 | 5.02% | 24.89% | 5.02% | 24.92% | **5.00**% | 24.89% |

Table 5: Performance comparison of GMsFEM and GMsFEM-NO for 3D ($100 \times 100 \times 100$, $N_v = 729$).

| $N_{bf}$ | Dataset | GMsFEM | | GMsFEM-NO, $\mathcal{L}_{SAL}$ | |
| | | $L_2$ | $H_1$ | $L_2$ | $H_1$ |
|---|---|---|---|---|---|
| | Diffusion, 8 | 1.62% | 14.76% | 1.68% | 14.98% |
| 8 | Diffusion, 9 | 3.0% | 12.55% | 3.04% | 12.84% |
| | Richards, 8 | 2.54% | 17.83% | 2.57% | 17.91% |
| | Richards, 9 | 2.55% | 17.85% | 2.57% | 17.91% |

Table 6: Basis generation time: GMsFEM-NO vs. standard GMsFEM offline stage.

| Grid | $N_v$ | GMsFEM, sec. | GMsFEM-NO, sec. |
|---|---|---|---|
| $100 \times 100$ | 36 | 16.87 | **0.28** |
| $250 \times 250$ | 121 | 210.5 | **0.31** |
| $50 \times 50 \times 50$ | 216 | 935.4 | **0.84** |
| $100 \times 100 \times 100$ | 729 | 10547.2 | **1.33** |

## 3.3 STANDALONE NOS VS. GMsFEM-NO

In this section, we compare GMsFEM-NO against standalone SOTA NOs, including F-FNO, GNOT Hao et al. (2023) and Transolver++ Luo et al. (2025). While these standalone models offer fast inference, their accuracy deteriorates significantly on high-contrast datasets. As shown in Table 7 (250×250 grid), GMsFEM-NO achieves superior accuracy; for Richards' equation with a complex source term (9), it reduces the relative L2 error by 2.8× compared to the best standalone NO. Full training details are in Appendix I.

A critical advantage of GMsFEM-NO over the standalone NO lies in its independence from the right-hand side terms of the PDE. The standalone NO exhibits catastrophic failure when tested on out-of-distribution forcing terms, as evidenced in Table 13 in Appendix K.

Since each coefficient contains multiple local domains of each type, GMsFEM-NO requires fewer samples than F-FNO for training. As shown in Table 8, when $N_{\text{train}}$ is reduced below 800, the error for F-FNO begins to increase significantly. In contrast, GMsFEM-NO's accuracy remains stable across the range of 800 to 400 samples. Even with only 200 samples, the performance degradation for GMsFEM-NO remains small; for example, on Richards' equation with the simple right-hand side (8), the error increases only modestly from 1.85% to 2.07%.

Table 7: Performance comparison of NOs and GMsFEM-NO ($250 \times 250$ grid)

| $N_{\text{bf}}$ | Dataset | F-FNO | GNOT | Transolver++ | GMsFEM-NO |
|---|---|---|---|---|---|
| | Diffusion, 8 | **1.02**% | 1.26% | 1.15% | 1.05% |
| 8 | Diffusion, 9 | 4.51% | 14.29% | 6.63% | **1.60**% |
| | Richards, 8 | 2.44% | 2.34% | 2.17% | **1.72**% |
| | Richards, 9 | 4.45% | 14.69% | 8.82% | **1.60**% |

Table 8: Comparison of F-FNO and GMsFEM-NO performance across different training dataset sizes for $250 \times 250$.

| $\mathcal{D}_{\text{train}}$ | | Diffusion, 8 | Diffusion, 9 | Richards, 8 | Richards, 9 |
|---|---|---|---|---|---|
| 200 | GMsFEM-NO | **1.33**% | **1.77**% | **2.07**% | **1.77**% |
| | F-FNO | 2.85% | 11.56% | 6.52% | 11.49% |
| 400 | GMsFEM-NO | **1.15**% | **1.63**% | **1.85**% | **1.62**% |
| | F-FNO | 1.60% | 8.16% | 4.17% | 8.41% |
| 600 | GMsFEM-NO | **1.12**% | **1.61**% | **1.78**% | **1.61**% |
| | F-FNO | 1.21% | 4.92% | 3.27% | 5.43% |
| 800 | GMsFEM-NO | 1.13% | **1.62**% | **1.82**% | **1.62**% |
| | F-FNO | **1.02**% | 4.51% | 2.44% | 4.45% |

## 3.4 GMsFEM-NO FOR DIFFERENT GRIDS

Table 9 demonstrates the resolution invariance of GMsFEM-NO by training the model on a coarse grid and testing it on a finer grid ($500 \times 500$), with results compared against the standard GMsFEM solution computed directly on the fine grid. We used a $10 \times 10$ coarse grid (121 subdomains) for all experiments. The results demonstrate the stability of the proposed method. GMsFEM-NO performs effectively when evaluated on a grid resolution higher than its training resolution, a key advantage enabled by the neural operator's ability to generalize to different discretizations.

Table 9: Evaluation of GMsFEM-NO trained on coarse grid and tested on finer grid, with comparison to standard GMsFEM.

| Train grid | Test grid | Diffusion, 8 | Diffusion, 9 | Richards, 8 | Richards, 9 |
|---|---|---|---|---|---|
| | | | GMsFEM-NO | | |
| 100 | 500 | 2.42% | 2.97% | 4.70% | 3.49% |
| 250 | 500 | 1.45% | 1.79% | 2.25% | 1.97% |
| | | | GMsFEM | | |
| | 500 | 1.17% | 1.46% | 1.93% | 1.66% |

## 4 CONCLUSION

In this work, we propose GMsFEM-NO, a novel method for solving multiscale PDEs that employs NOs to predict the multiscale basis function subspaces in the GMsFEM offline stage, replacing the conventional solution of local eigenvalue problems. We validated the method on standard 2D and 3D benchmarks: a linear elliptic diffusion problem and the nonlinear steady-state Richards equation. Additionally, we demonstrated its efficacy for time-dependent equations and problems with mixed boundary conditions in 2D. GMsFEM-NO achieves more than $60\times$ speedup in basis generation compared to standard GMsFEM.

A key contribution is a novel subspace alignment loss function, which enables direct learning of the basis function subspace and improves the $L_2$ accuracy over conventional $\mathcal{L}_{\text{RBFL}_2}$ loss. The GMsFEM-NO framework remains independent of the PDE's right-hand side, allowing it to maintain consistent performance across varying forcing terms. This contrasts with standalone NOs, which exhibit errors exceeding 100% on out-of-distribution data. Furthermore, GMsFEM-NO demonstrates greater data efficiency, requiring half the training samples of a comparable NO. A significant advantage is the method's discretization invariance: GMsFEM-NO performs effectively when evaluated on grid resolutions higher than those used for training, demonstrating strong generalization across different computational meshes. By preserving the mathematical structure of multiscale methods while leveraging NO speed, this work establishes a practical paradigm for heterogeneous PDE simulation.

The primary limitation of our method is its current restriction to structured grids due to the chosen NO architecture. Additionally, our experiments focused on relatively small grid sizes, which may not fully represent large-scale applications. While we successfully tested our approach on time-dependent equations and problems with mixed boundary conditions in 2D, the study was primarily focused on steady-state problems with Dirichlet boundary conditions.

Future work will expand this framework in several key directions. First, we will target more complex PDEs. Second, extending the framework to irregular domains is a critical next step. This is well-supported theoretically, as the GMsFEM methodology is established for unstructured meshes, and can be integrated with geometry-aware NOs like Transolver++. Alongside these goals, we will also investigate performance on finer grid resolutions and the impact of coarse-grid sizing to fully realize the method's potential for large-scale, real-world simulations.

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

## A  COARSE GRID

The notation $\omega_i$ refers to the $i$-th local domain, where the index corresponds to the numbering of points on the coarse grid. Fig. 2 shows examples of local domains $\omega_0$, $\omega_{20}$, and $\omega_{34}$, representing the **full**, **half**, and **corner** types in 2D. In 3D, there are four types: **full** (8 cells), **half** (4 cells), **quarter** (2 cells), and **corner** (1 cell), where the cell is a cube. Each local domain is discretized with a fine grid.

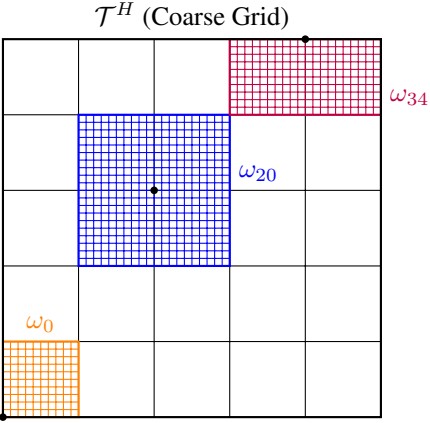

Figure 2: Illustration of a $5 \times 5$ coarse grid $\mathcal{T}_H$ showing local domains of different types: the **corner** type $\omega_0$ (1 cell), **half** type $\omega_{34}$ (2 cells), and **full** type $\omega_{20}$ (4 cells), where the cell is a square.

## B  TRAINING GMSFEM-NO

We train separate specialized NOs for each geometric domain type: three for 2D problems (full, half, corner) and four for 3D problems (full, half, quarter, corner), as illustrated for the 2D case in Fig. 3 (**a-c**). Each NO predicts the $N_{\text{bf}}$ basis functions for all local domains of its assigned type.

The number of local domains for each geometric type can be calculated based on the coarse grid dimensions. For a 2D grid with 36 domains ($5 \times 5$ cells), the counts are: 16 full, 16 half, and 4 corner

domains. For a finer 2D grid with 121 domains ($10 \times 10$ cells), the counts are: 81 full, 36 half, and 4 corner domains. In 3D, for a grid with 216 domains ($5 \times 5 \times 5$ cells), the distribution is: 64 full, 96 half, 48 quarter, and 8 corner domains. In 3D, for a finer grid with 729 domains ($8 \times 8 \times 8$ cells), the distribution is: 343 full, 294 half, 84 quarter, and 8 corner domains.

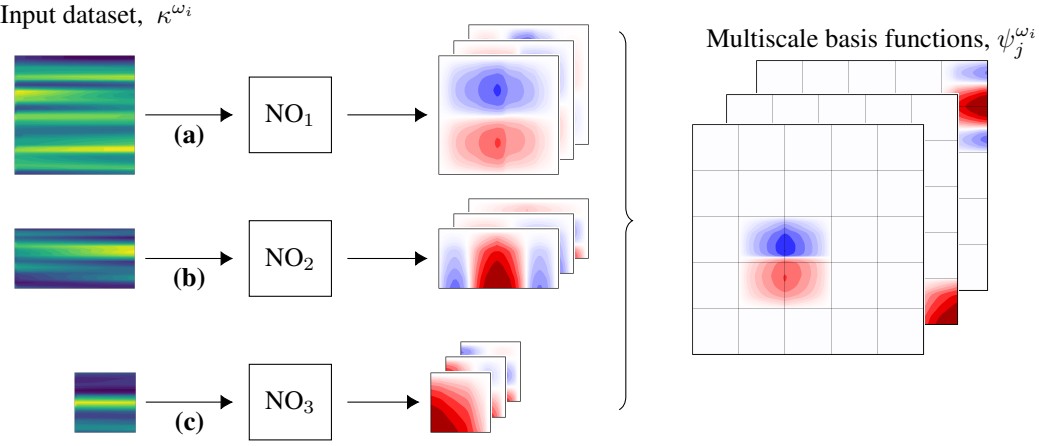

Figure 3: Multiscale basis generation algorithm for three subdomain types $\omega_i$: **(a) full**, **(b) half**, **(c) corner** - using dedicated NOs per type with further extension to $\Omega$.

## C    SUBSPACE ALIGNMENT LOSS (SAL)

To understand the relationship between the proposed $\mathcal{L}_{\text{SAL}}$ (5) and classical Grassmannian geometry Bendokat et al. (2024); Mandolesi (2023), we begin with the orthogonal projection matrices. For a subspace $R$ spanned by a set of basis vectors, we compute an orthonormal basis $Q_R$ via the thin QR decomposition. The orthogonal projection matrix onto $R$ is then given by $P_R = Q_R Q_R^\top$.

The Grassmannian distance between two $k$-dimensional subspaces $R$ and $\widetilde{R}$ is defined using these projection matrices. The distance derivation proceeds as follows:

$$\left\| P_R - P_{\widetilde{R}} \right\|_F^2 = \text{tr}(P_R) - 2\text{tr}(P_R P_{\widetilde{R}}) + \text{tr}(P_{\widetilde{R}}) = \dim(R) + \dim(\widetilde{R}) - 2\text{tr}(P_R P_{\widetilde{R}})$$
$$= 2\left(k - \text{tr}(Q_R Q_R^\top Q_{\widetilde{R}} Q_{\widetilde{R}}^\top)\right).$$

The matrix $Q_R^\top Q_{\widetilde{R}}$ contains the cosines of the principal angles between the subspaces. Therefore,

$$\left\| P_R - P_{\widetilde{R}} \right\|_F^2 = 2\left(k - \left\| Q_R^\top Q_{\widetilde{R}} \right\|_F^2\right).$$

Consequently, the Grassmannian distance simplifies to:

$$d(R, \widetilde{R}) = \frac{1}{\sqrt{2}} \left\| P_R - P_{\widetilde{R}} \right\|_F = \sqrt{k - \left\| Q_R^\top Q_{\widetilde{R}} \right\|_F^2}.$$

This derivation confirms that minimizing $\mathcal{L}_{\text{SAL}}$ is equivalent to minimizing the expected Grassmannian distance between the true and predicted subspaces.

In considering the theoretical soundness of our approach, we note that a formal upper bound linking principal angles to the final error would further strengthen the framework. When the SAL loss is small, it ensures that the learned solution remains close to the exact solution. To formalize this, let $Q_R c$ represent the multiscale solution computed using GMsFEM and $Q_{\widetilde{R}} c$ denote the learned

solution. The error can then be bounded as follows:

$$
\begin{aligned}
\|u - Q_{\tilde{R}}c\|^2 \leq \|u - Q_R c\|^2 + \|Q_R c - Q_{\tilde{R}}c\|^2 = \\
\|u - Q_R c\|^2 + c^\top (Q_R - Q_{\tilde{R}})^\top (Q_R - Q_{\tilde{R}})c = \\
\|u - Q_R c\|^2 + 2c^\top (I - (Q_R)^\top Q_{\tilde{R}})c.
\end{aligned}
\tag{10}
$$

The first term is GMsFEM error (which is small), and the second term is small if $I - (Q_R)^\top Q_{\tilde{R}}$ is small. The norm above can play an important role in the proof. Since the GMsFEM error is done in energy norm, in general, one needs to take the energy norm, which is non-local and can slow down computations.

This error estimate gives a bound between the learned solution and the angle between the spaces. To estimate the angle via the solution error is more difficult. Indeed, these questions need to be addressed in the future and will help to choose the appropriate loss functions.

## D $\quad \mathcal{L}_{\text{SAL-PR}}$

The primary role of $\mathcal{L}_{\text{SAL}}$ (5) is to enforce geometric alignment between the learned subspace $\widetilde{R}^i$ and the target GMsFEM subspace $R^i$. However, this loss has a specific limitation: it is invariant to rotations within the subspace. If $Q_{\widetilde{R}^i} = Q_{R^i} U$ for any unitary matrix $U$ (s.t. $U^\top U = I$), the subspaces are considered identical under this metric, as:

$$
N_{\text{bf}} - \left\| Q_{R^i}^\top Q_{\widetilde{R}^i} \right\|_F^2 = N_{\text{bf}} - \left\| U \right\|_F^2 = 0
$$

This geometric alignment may overlook finer discrepancies in how specific functions are projected. The $\mathcal{L}_{\text{SAL-PR}}$ (6) term was introduced to address this theoretical gap by directly enforcing projection consistency. It tests the subspaces with random vectors $v^i$ drawn from the target subspace $R^i$. Since $v^i$ lies in $R^i$, its projection via the true basis is itself: $P_{R^i} v^i = v^i$. The discrepancy $(I - P_{\widetilde{R}^i})v^i$ thus represents the projection error. Minimizing this forces the learned subspace to correctly capture arbitrary vectors from the target subspace, going beyond mere geometric overlap.

## E    THE SIGN INVARIANCE OF BASIS FUNCTIONS

Multiscale basis functions $\psi_j^{\omega_i}$ are derived from local eigenvectors $\phi_j^{\omega_i}$ via $\psi_j^{\omega_i} = \chi_i \phi_j^{\omega_i}$, where $\chi_i$ is a partition of unity function and $\phi_j^{\omega_i}$ solve the symmetric generalized eigenvalue problem:

$$
A_h^{\omega_i} \phi = \lambda S_h^{\omega_i} \phi,
$$

where $A_h^{\omega_i}$ and $S_h^{\omega_i}$ are symmetric matrices. These eigenvectors satisfy the orthogonality relations:

$$
\phi_j^\top A_h^{\omega_i} \phi_k = \lambda_j \delta_{jk}, \quad \phi_j^\top S_h^{\omega_i} \phi_k = \delta_{jk}.
$$

Since eigenvectors are defined only up to a scalar multiple, if $\phi_j$ is an eigenvector corresponding to eigenvalue $\lambda_j$, then $-\phi_j$ is also a valid eigenvector for the same eigenvalue. Both choices satisfy the orthogonality and normalization conditions above, meaning the sign of each basis function is arbitrary and does not affect its mathematical properties.

To address this sign ambiguity, we define the RBFL$_2$ loss as:

$$
\mathcal{L}_{\text{RBFL}_2} = \mathbb{E}_{i,j} \left[ \min \left( \frac{\|\psi_j^i - \tilde{\psi}_j^i\|_2^2}{\|\psi_j^i\|_2^2}, \frac{\|\psi_j^i + \tilde{\psi}_j^i\|_2^2}{\|\psi_j^i\|_2^2} \right) \right],
$$

where $\psi_j^i$ are the final multiscale basis functions (typically obtained by multiplying eigenvectors by partition of unity functions). This loss compares the prediction $\tilde{\psi}_j^i$ against both $\psi_j^i$ and $-\psi_j^i$, selecting the smaller error to account for the sign invariance.

## F  INPUT DATA

We use the Karhunen-Loève expansion (KLE) Wong (1971); Aarnes & Efendiev (2008); Vasilyeva et al. (2021) to generate stochastic permeability fields. This method decomposes a random field into deterministic spatial functions and random coefficients.

**1. Covariance Function**. We assume the covariance function has an exponential form:

$$R(x, \ y) = \sigma_R^2 \exp(-\sqrt{\Delta^2}),$$

with

$$\Delta^2 = \frac{|x_1 - x_2|^2}{l_x^2} + \frac{|y_1 - y_2|^2}{l_y^2},$$

for 2D case and

$$\Delta^2 = \frac{|x_1 - x_2|^2}{l_x^2} + \frac{|y_1 - y_2|^2}{l_y^2} + \frac{|z_1 - z_2|^2}{l_z^2},$$

for 3D case with correlation lengths $l_x, l_y, l_z$ and variance $\sigma_R^2$:

- For the 2D case: $l_x = 0.02, \ l_y = 0.6, \ \sigma_R^2 = 2$;
- For the 3D case: $l_x = 0.02, \ l_y = 0.6, \ l_z = 0.2, \ \sigma_R^2 = 2$.

**2. Eigenvalue Problem**. The eigenfunctions $\phi_k$ and eigenvalues $\lambda_k$ are obtained by solving the homogeneous Fredholm integral equation:

$$\int_\Omega R(x,y)\phi_k(y)dy = \lambda_k\phi_k(x), \quad k = 1, 2, \ldots,$$

**3. Random Field Construction**. The random field is represented as:

$$Y_L(x,\omega) = \sum_{k=1}^{L} \sqrt{\lambda_k}\theta_k(\omega)\phi_k(x),$$

where $\theta_k(\omega)$ are scalar random variables, and $L$ is chosen to capture most of the field's energy by retaining the largest eigenvalues.

**4. Permeability Field Generation**. Each stochastic permeability field is defined as:

$$\kappa(x,\omega) = \exp(a_k \cdot \phi(x,\omega)),$$

where $\phi(x,\omega)$ represents the heterogeneous porosity field derived from $Y_L(x,\omega)$, and $a_k > 0$ is a scaling parameter that controls the contrast.

This KLE framework provides a systematic approach for generating realistic permeability fields with prescribed spatial correlation structures. An example of a 2D input coefficient field $\kappa(x)$ is shown in Fig. 4.

The **spatially variable forcing term** is defined by

$$f(x) \sim \gamma \cdot \mathcal{N}\Big(\alpha \cdot \big(I - \Delta\big)^{-\beta}\Big),$$

where $\mathcal{N}$ denotes a Gaussian random field. The parameters are set as follows:

- For the 2D case: $\gamma = 2000, \alpha = 1$, and $\beta = 0.5$;
- For the 3D case: $\gamma = 2000, \ \alpha = 2$, and $\beta = 1$.

An example of a 2D right-hand side $f(x)$ is shown in Fig. 4.

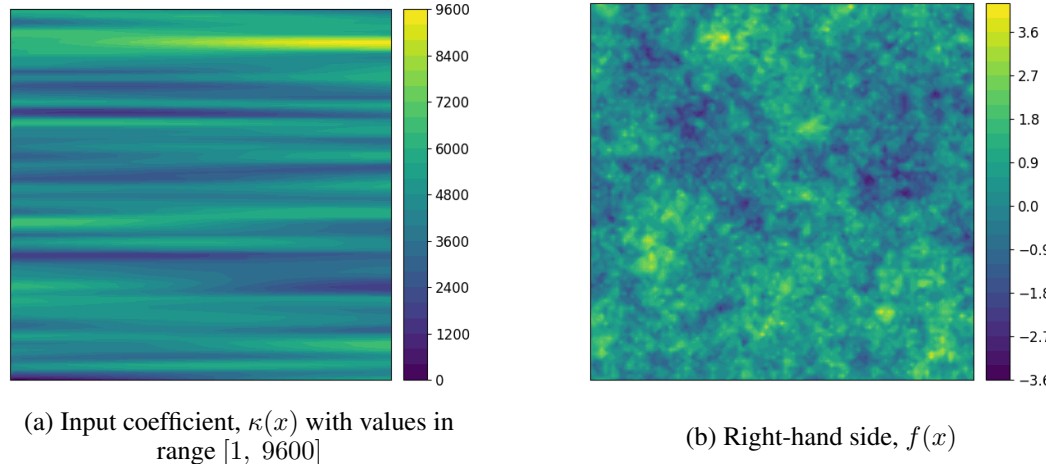

(a) Input coefficient, $\kappa(x)$ with values in range $[1, \ 9600]$

(b) Right-hand side, $f(x)$

Figure 4: Example of input coefficient and right-hand side.

# G    BASELINES VS. GMsFEM-NO

We compare GMsFEM-NO with several baselines:

1. **POD** Volkwein (2013). Classical global intrusive POD.
2. **Intrusive POD with DeepONet/FFNO or POD basis** Meuris et al. (2021), Meuris et al. (2023). First selected neural network is trained on standard regression problem. After that one extract basis from trained network and uses similar to intrusive POD to form reduced model. For **FFNO** Tran et al. (2021) basis is extracted from the last hidden layer, for **DeepONet** Lu et al. (2019) basis is extracted from trunk net.
3. **PCA-Net** Hesthaven & Ubbiali (2018), Bhattacharya et al. (2021). POD is used to compress features and targets, MLP is used as processor.
4. **Kernel** Batlle et al. (2024). Vector RKHS method is used to map sampled input functions to sampled output functions.
5. **DeepPOD** Franco et al.. A DL-based techniques used to directly learn optimal basis with projector-based loss.

We use dataset with spatially variable forcing term (9) covered in more details in Appendix F. Neural networks was trained and evaluated on grid $100 \times 100$.

For each selected baseline we perform sweep over hyperparameters:

1. **Intrusive POD with FFNO basis.** Architecture is defined by the number of features in the hidden layer, number of modes used by spectral convolution, and number of layers. Number of features in the hidden layer was fixed to 64, number of modes was selected from the set $[10, 14, 16]$, number of layers – from the set $[3, 4, 5]$. Optimisation was performed with Lion optimiser Chen et al. (2023) with exponential decay 0.5 with number of transition steps selected from $[100, 200]$, and learning rate selected from $[5 \cdot 10^{-5}, 10^{-4}]$. We optimise for 1000 epoch with batch size 10. In all architectures we used GELU activation function.
2. **Intrusive POD with DeepONet basis.** Architecture is defined by trunk and branch nets. As trunk net we used convolution architecture with spatial downsampling by a factor of 2 along each dimension after each layer, simultaneously, the number of channels was multiplied by 2 after each layer, as branch net we used standard MLP. We apply optimisation similar to the one of FFNO, but select learning rate from $[10^{-3}, 10^{-4}]$. Number of trunk network layers was fixed to 4, trunk encoder transformed 2 input features to either 4 or 5 features, kernel size of convolution in trunk was selected among $[3, 7]$. In the branch net we vary number of layers $[3, 4]$ and the number of basis vectors $[100, 200]$ in the last layer.
3. **DeepPOD** Grid search for DeepPOD was exactly the same as for Intrusive POD with FFNO.

4. **PCANet.** For PCANet the optimisation was similar to Intrusive POD with DeepONet, but with 3000 epochs. We vary the sizes of POD encoder and decoder among $[100, 300, 500]$ and $[100, 300, 500]$. For MLP processor we vary the number of layers $[3, 4, 5]$ and the number of hidden neurons $[100, 300, 500]$.

5. **Kernel.** We closely followed code provided by authors. As kernels we used Matern, RBF. We combined the method with POD and performed a grid search over the number of modes: $[50, 100, 150, 200]$ for both features and targets.

Table 10: Regression-based methods.

| method | train error | test error |
|---|---|---|
| GMsFEM-NO | 2.6% | 2.8% |
| PCANet | 6% | 24% |
| kernel | 7% | 100% |

Comparison of regression-based approaches with GMsFEM-NO appears in Table 10. We observe significant overfitting for kernel-based method and PCA-Net.

Intrusive techniques are compared in Figure 5. We see that bases extracted from DeepONet and FFNO are generally not appealing. FFNO slightly improves over global POD (weak baseline) for $\simeq 64$ basis functions. DeepONet fails to reach accuracy of global POD. The most competative approach is DeepPOD. Note however, that DeepPOD becomes comparable to GMsFEM-NO with 8 sparse (localised) basis functions only when it uses 20 dense basis functions. With 30 basis functions DeepPOD outperforms GMsFEM-NO with 8 basis functions.

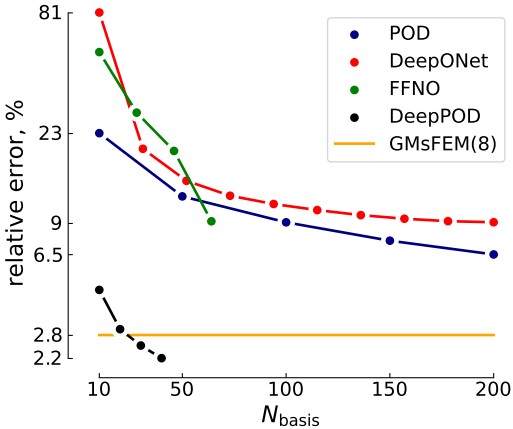

Figure 5: Comparison of accuracy for intrusive techniques. Number of basis functions for GMsFEM is fexed to 8. DeepONet and FFNO mean Intrusive POD with DeepONet/FFNO basis.

## H  GMsFEM-NO TRAINING DETAILS

For F-FNO training to predict basis functions subspace, we used AdamW optimizer Loshchilov & Hutter (2017) with cosine decay learning rate scheduler. The initial learning rate was $1 \cdot 10^{-3}$. We trained NO for 600 epochs. The best number of random test vectors $v^i$ is 10 (see the Table 11). For the rest of the hyperparameters, we performed a grid search:

1. Batch size $[8, 16, 32]$.

2. Number of operator layers $[4, 5]$.

3. Number of modes used in F-FNO kernel:
    - For 2D:

- **–** for full domains $\big[[16, 16], [18, 18]\big]$;
- **–** for half domains $\big[[8, 8], [10, 10], [14, 8], [14, 10]\big]$;
- **–** for corner domains $\big[[6, 6], [8, 8], [10, 10]\big]$.
- • For 3D:
  - **–** for full domains $\big[[6, 6, 6], [8, 8, 8]\big]$;
  - **–** for half domains $\big[[8, 8, 4], [6, 6, 3]\big]$;
  - **–** for quarter domains $\big[[8, 4, 4], [6, 3, 3]\big]$.
  - **–** for corner domains $\big[[4, 4, 4], [3, 3, 3]\big]$.
4. Number of channels in the FFNO kernel $[64, 128]$.

Table 11: Performance GMsFEM-NO for 2D ($100 \times 100$, $N_v = 36$).

| Dataset | GMsFEM-NO, $\mathcal{L}_{\text{SAL-PR}}$ | | | | | | | |
| | $v^i = 5$ | | $v^i = 10$ | | $v^i = 15$ | | $v^i = 20$ | |
| | $L_2$ | $H_1$ | $L_2$ | $H_1$ | $L_2$ | $H_1$ | $L_2$ | $H_1$ |
| --- | --- | --- | --- | --- | --- | --- | --- | --- |
| Diffusion, 8 | 1.09% | 11.90% | **1.06**% | 11.65% | 1.31% | 12.34% | 1.10% | 12.13% |
| Diffusion, 9 | 2.84% | 19.14% | **2.81**% | 19.03% | 2.88% | 19.38% | 2.87% | 19.37% |
| Richards, 8 | 1.91% | 11.21% | **1.87**% | 11.25% | 1.98% | 11.75% | 1.94% | 11.64% |
| Richards, 9 | **2.91**% | 20.06% | 2.99% | 19.60% | 2.96% | 20.33% | 2.94% | 20.26% |

The source code containing the optimal parameters will be made publicly available upon acceptance. We use JAX, Optax DeepMind et al. (2020) and Equinox Kidger & Garcia (2021) in all experiments.

# I  F-FNO, GNOT, TRANSOLVER++ TRAINING DETAILS

## I.1  F-FNO TRAINING DETAILS

For F-FNO Tran et al. (2021), we used the following training protocol. We employed the AdamW optimizer Loshchilov & Hutter (2017) with a cosine decay learning rate scheduler and trained for 600 epochs, using a base learning rate of $10^{-3}$. We performed a grid search over the following hyperparameters:

1. Batch size: $[8, 16, 32]$;
2. Number of modes in F-FNO kernel: $[14, 16]$;
3. Number of operator layers: $[4, 5]$;
4. Number of channels in F-FNO kernel: $[64, 128]$.

The optimal hyperparameters were: batch size 8, 5 operator layers, 16 modes, and 128 channels.

## I.2  GNOT TRAINING DETAILS

For GNOT Hao et al. (2023), we employed the following training protocol: AdamW optimizer with a onecycle learning rate scheduler for 600 epochs, using a base learning rate of $10^{-3}$. We conducted a grid search over hyperparameters that were previously found optimal for different 2D problems in Hao et al. (2023):

1. Batch size: $[4, 8]$
2. Number of attention layers: $[3, 4]$
3. Hidden size of attention and input embeddings: $[96, 128, 192]$
4. Number of MLP layers: $[3, 4]$

5. Hidden size of MLP: $[128, 192]$

6. Number of heads: $[4, 8]$

7. Number of experts: $[3, 4]$

The optimal configuration was: batch size 4, 4 attention layers, hidden size of 128 for attention, MLP, and input embeddings, 8 heads, 4 experts, and 4 MLP layers.

### I.3    TRANSOLVER++ TRAINING DETAILS

For Transolver++ Luo et al. (2025), we employed the following training protocol: AdamW optimizer with a onecycle learning rate scheduler for 600 epochs, using a base learning rate of $10^{-3}$. We conducted a grid search over hyperparameters previously identified as optimal for 2D problems in Luo et al. (2025):

1. Batch size: $[4, 8]$

2. Number of layers: $[4, 8]$

3. Hidden size: $[128, 256]$

4. Number of MLP layers: $[1, 2]$

5. Number of heads: $[4, 8]$

6. Slices: $[32, 64]$

The optimal configuration was: batch size 4, 8 layers, hidden size 256, 2 MLP layers, 8 heads, and 64 slices.

## J    RESULTS OF GMsFEM-NO FOR $N_{\mathrm{BF}} = 4$

Table 12: Performance comparison of loss functions for NO training ($100 \times 100$ grid, $N_v = 36$).

| $N_{\mathrm{bf}}$ | Dataset | $\mathcal{L}_{\mathrm{RBFL}_2}$ | | $\mathcal{L}_{\mathrm{SAL}}$ | | $\mathcal{L}_{\mathrm{SAL\text{-}PR}}$ | |
|---|---|---|---|---|---|---|---|
| | | $L_2$ | $H_1$ | $L_2$ | $H_1$ | $L_2$ | $H_1$ |
| | Diffusion, 8 | 2.72% | 19.10% | **2.39**% | 18.01% | 2.40% | 18.01% |
| 4 | Diffusion, 9 | 6.03% | 29.10% | **5.76**% | 28.40% | 5.78% | 28.43% |
| | Richards, 8 | 3.87% | 16.26% | **3.14**% | 15.01% | 3.17% | 15.02% |
| | Richards, 9 | 9.78% | 34.39% | **6.11**% | 29.37% | 6.13% | 29.42% |

For the Richards equation with complex right-hand side (9) and $N_{\mathrm{bf}} = 4$ basis functions, our proposed loss improves the relative $L_2$ metric by a factor of 1.6.

We did not conduct further experiments with GMsFEM-NO using $N_{\mathrm{bf}} = 4$ because its performance was insufficient and it underperformed compared to the standalone neural operator.

## K    OUT-OF-DISTRIBUTION RESULTS

Unlike standalone NOs, which suffer from catastrophic failure when applied to out-of-distribution forcing terms (Table 13), GMsFEM-NO is fundamentally independent of the right-hand side, ensuring robust performance. Retraining the NO for new right-hand side terms requires computationally expensive recalculation of solutions, highlighting a key limitation of standalone NO learning.

Table 13: Out-of-distribution results for the NO: training and testing on PDEs with different right-hand sides.

| Train, $\mathcal{D}_{\text{train}}$ | Test, $\mathcal{D}_{\text{test}}$ | $100 \times 100$ | $250 \times 250$ |
|---|---|---|---|
| Diffusion, 8 | Diffusion, 9 | 218% | 174% |
| Diffusion, 9 | Diffusion, 8 | 1392% | 1632% |
| Richards, 8 | Richards, 9 | 196% | 113% |
| Richards, 9 | Richards, 8 | 6503% | 6554% |

## L HEAT EQUATION FOR 2D

We consider the heat equation with heterogeneous coefficient

$$\frac{\partial u}{\partial t} - \nabla \cdot \big(\kappa(x)\nabla u(x)\big) = f(x), \quad x \in \Omega \times \big(0,\, T_{\max}\big],$$
$$u = 0, \quad x \in \partial\Omega \times \big(0,\, T_{\max}\big],$$
$$u\big|_{t=0} = 0, \quad x \in \Omega,$$

where time parameters: $\Delta t = 2.5 \cdot 10^{-6}$, $T_{\max} = 5 \cdot 10^{-4}$. We consider two right-hand side configurations:

- Uniform unit forcing term
$$f(x) = 1; \tag{11}$$

- Spatially variable forcing defined by
$$f(x) = \sin(\pi x)\cos(\pi y). \tag{12}$$

Table 14: Performance comparison of GMsFEM and GMsFEM-NO for Heat equation for 2D ($250 \times 250$, $N_v = 121$).

| $N_{\text{bf}}$ | Dataset | GMsFEM | | GMsFEM-NO, $\mathcal{L}_{\text{SAL}}$ | |
|---|---|---|---|---|---|
| | | $L_2$ | $H_1$ | $L_2$ | $H_1$ |
| 8 | Heat, 11 | 0.72% | 10.81% | 0.78% | 13.06% |
| | Heat, 12 | 1.13% | 19.92% | 1.17% | 21.57% |

The results, presented in the Table 14, show that our proposed GMsFEM-NO method with $\mathcal{L}_{\text{SAL}}$ maintains high accuracy for this time-dependent problem, with relative $L_2$ errors below 1.2%. This demonstrates the successful application of our framework to time-dependent problems.

## M DIFFUSION EQUATION WITH MIXED BOUNDARY CONDITIONS

We consider the diffusion equation with heterogeneous coefficient

$$-\nabla \cdot \big(\kappa(x)\nabla u(x)\big) = f(x), \quad x \in \Omega \equiv (0,\, 1)^2,$$
$$u = 0, \quad x \in \Gamma_D,$$
$$\kappa(x)\frac{\partial u}{\partial n} = \alpha \cdot \big(u - u_{\text{const}}\big), \quad x \in \Gamma_R,$$

where the boundary is partitioned into a Dirichlet part $\Gamma_D = \big\{x \in \partial\Omega \mid y = 0\big\}$ and a Robin part $\Gamma_R = \big\{x \in \partial\Omega \mid y = 1\big\}$ with parameters: Robin coefficient $\alpha = 1$, $u_{\text{const}} = 1$.

Our framework demonstrates robust performance when extended to problems with mixed boundary conditions, maintaining high accuracy even in these more complex scenarios. As shown in Table 15, the GMsFEM-NO method with $\mathcal{L}_{\text{SAL}}$ achieves relative $L_2$ errors of approximately 1.2% for diffusion problems with Robin boundary conditions. This represents a significant extension beyond the homogeneous Dirichlet conditions typically considered in multiscale methods.

Table 15: Performance comparison of GMsFEM and GMsFEM-NO for equation with mixed boundary conditions for 2D ($250 \times 250$, $N_v = 121$).

| $N_{bf}$ | Dataset | GMsFEM | | GMsFEM-NO, $\mathcal{L}_{SAL}$ | |
|---|---|---|---|---|---|
| | | $L_2$ | $H_1$ | $L_2$ | $H_1$ |
| 8 | Diffusion, 8 | 1.11% | 7.98% | 1.22% | 12.10% |
| | Diffusion, 9 | 1.09% | 7.72% | 1.23% | 12.31% |

## N  TIME

The core objective of GMsFEM-NO is to reduce the cost of the traditional GMsFEM offline stage over many subsequent simulations. To provide full transparency, we have performed a detailed analysis that compares two practical scenarios: training models sequentially (one after another on one GPU) and in parallel. The breakeven point $x$ is the number of inference samples where the total time of GMsFEM-NO (including training data generation and training) equals that of using standard GMsFEM for all samples:

$$T_{data} + T_{train} + T_{inf} \cdot x = T_{GMsFEM} \cdot x,$$

where $T_{data}$ is the wall-clock time for generating training data via traditional GMsFEM, $T_{train}$ is the total wall-clock time for training NO (which varies by scenario), $T_{GMsFEM}$ is the wall-clock time for a single offline GMsFEM basis generation, and $T_{inf}$ is the wall-clock time for a single GMsFEM-NO inference. The results are in the Table 16.

Even with sequential training, the method pays off for multi-query scenarios. The benefit is larger for large-scale 3D problems, where the breakeven point remains low (159 samples) due to the high cost of the traditional GMsFEM solver.

Table 16: Performance comparison of sequential and parallel implementations

| Problem Configuration | Performance Metrics | | | | | |
|---|---|---|---|---|---|---|
| | $T_{GMsFEM}$ | $T_{inf}$ | Samples | Implementation | $T_{train}$ | $x$ |
| $100 \times 100$ | 16.87 | 0.28 | 800 | Sequential | 37800 | $-3270$ |
| | 16.87 | 0.28 | 800 | Parallel | 12600 | $-1570$ |
| $250 \times 250$ | 210.5 | 0.31 | 800 | Sequential | 43200 | $-1040$ |
| | 210.5 | 0.31 | 800 | Parallel | 14400 | $-870$ |
| $50 \times 50 \times 50$ | 935.4 | 0.84 | 800 | Sequential | 86400 | $-1090$ |
| | 935.4 | 0.84 | 800 | Parallel | 21600 | $-820$ |
| $100 \times 100 \times 100$ | 10547.2 | 1.33 | 150 | Sequential | 100800 | $-159$ |
| | 10547.2 | 1.33 | 150 | Parallel | 25200 | $-152$ |

