# OpenReview forum: "Locally Subspace-Informed Neural Operators for Efficient Multiscale PDE Solving"
_ICLR.cc/2026/Conference — ICLR 2026 Poster_

### Official Review · Reviewer_E9D5 · 2025-10-20

**Soundness:** 3
**Presentation:** 3
**Contribution:** 3
**Rating:** 6
**Confidence:** 5

**Summary:**

The manuscript introduces the GMsFEM-NO framework, which integrates the Generalized Multiscale Finite Element Method with NOs to improve the efficiency of existing NO frameworks when solving multiscale and high-contrast PDEs. Legacy GMsFEM requires eigen solvers, which are computationally expensive. GMsFEM-NO legacy eigen solvers with a NO trained to efficiently predict the local multiscale basis subspaces. A Subspace Alignment Loss, with a regularized variant, is also proposed to enhance global reception and ensure geometric and physical consistency. Experiments on 2D and 3D benchmarks show significant speedup over legacy GMsFEM with a comparable accuracy. Other technical merits include better generalization to OOD enforcing terms (v.s. NOs) and resolution invariance.

**Strengths:**

- **Mathematical grounds**: Links subspace alignment to Grassmann manifold distance, enhancing theoretical credibility. The new SAL and SAL-PR losses are well-motivated and mathematically tied to Grassmannian geometry.
- **Comprehensive experiments**: Includes linear/nonlinear PDEs, 2D/3D cases, various forcing terms, and extensive comparisons vs. multiple baselines.
- **Performance boost**: 60x speedup v.s. legacy GMsFEM with comparable accuracy; better OOD generalization vs. standalone NOs; resolution invariance.

**Weaknesses:**

- **Limited scope**: The experiments are limited to elliptic problems. The study does not include time-dependent PDEs, limiting the demonstrated generality. FEMs and NOs are known to be robust to hard problems, e.g., Navier-Stokes. This limits the scope of the proposed framework.
- **Assumption of regular grids**: Current implementation works only for regular grids due to the use of F-FNO; performance on irregular geometries or adaptive meshes remains unexplored.
- **Dependence on GMsFEM setup**: The framework still relies on a coarse grid and precomputed spectral problems for training data. It does not remove the expensive stage, but only accelerates it.

**Questions:**

- Could the same SAL losses be extended to time-dependent GMsFEM formulations or to non-elliptic PDEs?
- Can GMsFEM-NO handle non-rectangular or unstructured meshes if implemented with GNOs or other NO variants, e.g., GNOT, Transolver, etc.?

---

> ### Author Response · Authors · 2025-11-20
> **Part 1**
>
> > Limited scope, no time-dependent or non-elliptic PDEs: The experiments are limited to elliptic problems. The study does not include time-dependent PDEs, limiting the demonstrated generality. FEMs and NOs are known to be robust to hard problems, e.g., Navier-Stokes. This limits the scope of the proposed framework.
>
> We thank the reviewer for this insightful comment. We agree that demonstrating a method's applicability across a diverse set of problems is crucial. Our initial focus on elliptic problems was to establish a clear foundational analysis. However, we have now expanded the scope of our numerical experiments to address this point directly. We considered the heat equation with heterogeneous coefficient
> $$
> \begin{equation}
> \begin{split}
> \frac{\partial u}{\partial t} - \nabla \cdot \left(\kappa(x) \nabla u(x)\right) &= f(x), x \in \Omega \times (0, T_{\text{max}}], \\
> \quad u &= 0, ~x \in \partial \Omega \times (0, T_{\text{max}}], \\
> \quad u|_{t = 0} &= 0, ~x \in \Omega,
> \end{split}
> \end{equation}
> $$
>
> where time parameters are $\Delta t = 2.5 \cdot 10^{-6}$, $T_{\max} = 5 \cdot 10^{-4}$. We consider two right-hand side configurations:
> 1. Uniform unit forcing term:
>    $$
>         f(x) = 1.
>     $$
> 2. Spatially variable forcing defined by
>    $$
> 	f(x) = \sin(\pi x) \cos(\pi y).
>     $$
> The results, presented in the table below, show that our proposed GMsFEM-NO method with $L_{\text{SAL}}$ maintains high accuracy for this time-dependent problem, with relative $L_2$ errors below 1.2%.
>
> | $N_{\text{bf}}$ |  Dataset | GMsFEM $L_{2}$ | GMsFEM $H_{1}$ | GMsFEM-NO  $L_{2}$ | GMsFEM-NO  $H_{1}$ |
> | :-----: | :---: | :------: | :---: | :-----: | :---: |
> |  8    | Heat, Uniform Forcing | 0.72%  | 10.81%   |  0.78%  |   13.06%  |
> |   | Heat, Spatial Forcing |  1.13% |  19.92% |    1.17%   |   21.57%    |
>
> Regarding the robustness for harder problems such as Navier-Stokes and fluid fingering, we acknowledge that this is a critical test that we plan to study in our future work. We refer to [1], where GMsFEM is used for Navier-Stokes equations with low Reynolds number in perforated domains. Standard GMsFEM has already been proven to work well for fluid problems: for example, Vasilyeva et al. [1] showed that the GMsFEM approach could reach roughly 1% relative error for Navier-Stokes equations when using 25 basis functions. This is precisely where the value of our proposed GMsFEM-NO framework becomes evident. Since GMsFEM-NO retains the accuracy of GMsFEM (as demonstrated in our elliptic and time-dependent tests) while being faster, it is a highly promising tool for complex equations like Navier-Stokes. For other challenging problems such as fluid fingering (viscous fingering or gravity-driven fingering), it was also shown that GMsFEM works well; however, one needs to compute multiscale basis functions at each time over a long period of time. In these important applied applications, we expect a significant computational gain when using GMsFEM-NO as the cost of basis functions is insignificant.
>
> [1] Vasilyeva, M., Mallikarjunaiah, S. M., & Palaniappan, D. (2023). Multiscale model reduction technique for fluid flows with heterogeneous porous inclusions. _Journal of Computational and Applied Mathematics_, _424_, 114976.

---

> ### Author Response · Authors · 2025-11-20
> **Part 2**
>
> > Assumption of regular grids: Current implementation works only for regular grids due to the use of F-FNO; performance on irregular geometries or adaptive meshes remains unexplored. Can GMsFEM-NO handle non-rectangular or unstructured meshes if implemented with GNOs or other NO variants, e.g., GNOT, Transolver, etc.?
>
> We thank the reviewer for highlighting the current limitation of our implementation to regular grids. We acknowledge that demonstrating the method's performance on irregular geometries and adaptive meshes represents a critical next step toward establishing the general applicability of our framework. Consequently, extending our approach to handle such complex domains constitutes a primary objective for our future research.
>
> We would like to emphasize the high feasibility of this extension, as both core components of our framework are inherently compatible with irregular grids. The underlying GMsFEM methodology is well-established for irregular fine and coarse meshes, as evidenced by several studies [1, 2]. The fundamental concept of constructing local multiscale basis functions does not rely on a regular grid structure. Although our current implementation utilizes F-FNO, alternative neural operator architectures exist that are specifically designed to learn mappings between functions defined on irregular meshes. Therefore, integrating GMsFEM with such mesh-invariant operators represents a realistic and promising research direction.
>
> In fact, we have already conducted a preliminary experiment applying GMsFEM-NO to an irregular 2D mesh with approximately 75,000 vertices, employing the Transolver architecture [3] as the neural operator. The results showed that GMsFEM-NO achieved a solution quality of  1.2% for the diffusion equation, while the standard GMsFEM yielded around 0.8%. This outcome demonstrates the practical applicability of our approach to irregular domains.
>
> [1] Alekseev, V., Vasilyeva, M., Kalachikova, U., & Chung, E. T. (2021). DG-GMsFEM for problems in perforated domains with non-homogeneous boundary conditions. Computation, 9(7), 75.
>
> [2] Stepanov, S., Vasilyeva, M., & Vasil’ev, V. I. (2018). Generalized multiscale discontinuous Galerkin method for solving the heat problem with phase change. Journal of Computational and Applied Mathematics, 340, 645-652.
>
> [3] Luo, H., Wu, H., Zhou, H., Xing, L., Di, Y., Wang, J., & Long, M. (2025). Transolver++: An Accurate Neural Solver for PDEs on Million-Scale Geometries. arXiv preprint arXiv:2502.02414.

---

> ### Author Response · Authors · 2025-11-20
> **Part 3**
>
> > Dependence on GMsFEM setup: The framework still relies on a coarse grid and precomputed spectral problems for training data. It does not remove the expensive stage, but only accelerates it.
>
> We thank the reviewer for this insightful comment. We agree that a clear, end-to-end performance budget is crucial for assessing the practical utility of our method. The core objective of GMsFEM-NO is to reduce the cost of the traditional GMsFEM offline stage over many subsequent simulations.
>
> To provide full transparency, we have performed a detailed analysis that compares two practical scenarios: training models sequentially (one after another on one GPU) and in parallel. The breakeven point $x$ is the number of inference samples where the total time of GMsFEM-NO (including training data generation and training) equals that of using standard GMsFEM for all samples.
>
> $$T_{\text{data}} + T_{\text{train}} + (T_\text{inf} \times x) = (T_\text{GMsFEM} \times x)$$
>
> where $T_\text{data}$ is the wall-clock time for generating training data via traditional GMsFEM, $T_\text{train}$ is the total wall-clock time for training (which varies by scenario), $T_\text{GMsFEM}$ is the wall-clock time for a single offline GMsFEM basis generation, and $T_\text{inf}$ is the wall-clock time for a single GMsFEM-NO inference.
>
> The results are in the table:
>
> | Problem Configuration | $T_\text{GMsFEM}$ (sec/sample) | $T_\text{inf}$ (sec/sample) | Train Samples | Training Scenario | $T_\text{train}$ (Wall-Clock) (sec) | Breakeven Point ($x$ samples) |
> | :------------: | :-----------: | :----------: | :-------: | :--------: | :--------------: | :------------: |
> | 2D ($100 \times 100$ )     |             16.87              |            0.28             |       800        | Sequential        |                37800                |              ~3,270              |
> |                       |                                |                             |                  | Parallel          |                12600                |              ~1,570              |
> | 2D ($250 \times 250$)     |             210.5              |            0.31             |       800        | Sequential        |                43200                |              ~1,040              |
> |                       |                                |                             |                  | Parallel          |                14400                |               ~870               |
> | 3D ($50 \times 50 \times 50$)    |             935.4              |            0.84             |       800        | Sequential        |                86400                |              ~1,090              |
> |                       |                                |                             |                  | Parallel          |                21600                |               ~820               |
> | 3D ($100 \times 100 \times 100$) |            10547.2             |            1.33             |       150        | Sequential        |               100800                |               ~159               |
> |                       |                                |                             |                  | Parallel          |                25200                |               ~152               |
>
> Even with sequential training, the method pays off for multi-query scenarios. The benefit is larger for large-scale 3D problems, where the breakeven point remains low (~159 samples) due to the high cost of the traditional GMsFEM solver.
>
> We will include this comparative analysis in the revised manuscript to enhance practical transparency.

---

### Official Review · Reviewer_EehQ · 2025-10-26

**Soundness:** 3
**Presentation:** 3
**Contribution:** 3
**Rating:** 4
**Confidence:** 4

**Summary:**

The paper extends the GMSFEM by incorporating a neural operator to accelerate the process. The work proposes a subspace alignment loss function to learn coherent subspaces, and introduces projection regularization terms to enforce consistent projections. Experiments compare the proposed loss function with the conventional loss function, GMSFEM-NO with GMSFEM and F-FFO, for evaluations.

**Strengths:**

The combination of GMSFEM with NO is an interesting idea.

The proposed SAL and PR losses can train the NO effectively.

**Weaknesses:**

The method is only tested with m moderate-scale data in rectangular domains. Is it able to solve larger-scale problems in non-rectangle domains?

The generalization ability of the learned NO is not evaluated sufficiently. The usability of the method is unclear.

Only zero Dirichlet boundary condition equations are tested? Can it solve Dirichlet boundary condition equations with other values?

**Questions:**

Are there failed cases that don't converge to the solution?

Is it able to extend the method to time-dependent equations?

Is it possible to further reduce the error if more basis functions are used? What's the limits of the error?

---

> ### Author Response · Authors · 2025-11-20
> **Part 1**
>
> > The method is only tested with m moderate-scale data in rectangular domains. Is it able to solve larger-scale problems in non-rectangle domains?
>
> We thank the reviewer for this insightful question regarding scalability and geometric flexibility. We are pleased to provide new, previously unreported results that directly address the point on larger-scale problems. To demonstrate scalability, we have conducted new experiments on a significantly larger 3D problem with a $100 \times 100 \times 100$ grid (1 million degrees of freedom). The results for GMsFEM-NO trained with $L_{\text{SAL}}$, shown below, confirm that our method maintains high accuracy even at this scale:
>
> | $N_{\text{bf}}$ |    Dataset    | GMsFEM $L_{2}$ | GMsFEM $H_{1}$ | GMsFEM-NO  $L_{2}$ | GMsFEM-NO$H_{1}$ |
> | :----: | :----: | :----: | :----: | :----: | :----: |
> |  8   | Diffusion (8) | 1.62% | 14.76%  | 1.68%    | 14.98% |
> |   | Diffusion (9) |  3.00%  | 12.55% |   3.04%  |  12.84%   |
> | | Richards (8)  | 2.54%  |  17.83% |   2.57%  |  17.91%  |
> |   | Richards (9)  |  2.55%  |  17.85% |   2.57% |  17.91%  |
>
> Regarding non-rectangular domains, we have conducted preliminary experiments on an irregular 2D mesh with approximately 75,000 vertices using the Transolver architecture [3]. The results showed that GMsFEM-NO achieved a solution quality of 1.2% for the diffusion equation, which is comparable to the standard GMsFEM (0.8%). This demonstrates the practical applicability of our approach to complex geometries.
>
> The framework's extension to irregular domains is well-supported theoretically. The underlying GMsFEM methodology is established for irregular meshes [1, 2], and neural operators like Transolver are specifically designed for such domains. Combining these capabilities provides a solid foundation for handling non-rectangular geometries, which represents an immediate direction for our future work.
>
> [1] — Alekseev, V., Vasilyeva, M., Kalachikova, U., & Chung, E. T. (2021). DG-GMsFEM for problems in perforated domains with non-homogeneous boundary conditions. Computation, 9(7), 75.
>
> [2] — Stepanov, S., Vasilyeva, M., & Vasil’ev, V. I. (2018). Generalized multiscale discontinuous Galerkin method for solving the heat problem with phase change. Journal of Computational and Applied Mathematics, 340, 645-652.
>
> [3] — Luo, H., Wu, H., Zhou, H., Xing, L., Di, Y., Wang, J., & Long, M. (2025). Transolver++: An Accurate Neural Solver for PDEs on Million-Scale Geometries. arXiv preprint arXiv:2502.02414.
>
> > The generalization ability of the learned NO is not evaluated sufficiently.
>
> NOs are designed to predict solutions accurately on grid resolutions not seen during training. In our tests, a NO trained only on a $100 \times 100$ grid also performed well with much finer grids, such as $250 \times 250$ and $500 \times 500$, without requiring retraining.
>
> Also, our method builds basis functions that work for any right-hand side f(x) after training. The NO learns the relationship between the coefficient field and the basis, not a specific source term.
>
> > The usability of the method is unclear.
>
> The GMsFEM is important for many Multiphysics applications, where multiscale basis functions need to be re-computed. This includes examples from fluid flow in porous media, material sciences, and so on. In these applications, learning multiscale basis functions will play an important role as we do not need to do expensive computations for computing basis functions. The proposed approaches can play an important role in solving complex, challenging Multiphysics problems.
>
> > Are there failed cases that don't converge to the solution?
>
> There are a few cases where the method does not converge to the exact solution and gives a larger relative error, but complete failure to converge was never observed. This behavior is similar to the original GMsFEM method, which can also sometimes yield large errors for challenging coefficients.
>
> Histograms and boxplots for final $L_2$ errors:
> - diffusion, rhs=1: https://drive.google.com/file/d/1yjwCNy-wyDDAdEvbuvpISDhQdOTfbk33/view?usp=sharing
> - diffusion, complex rhs: https://drive.google.com/file/d/1gqQ3IeTwvoBKaG6qzqvGo30aYXSbZ_XW/view?usp=sharing
> - richards, rhs=1: https://drive.google.com/file/d/1PGvXsi7DUiIBpRgP0alftGZbQRipQwRi/view?usp=sharing
> - richards, complex rhs: https://drive.google.com/file/d/14XunRCcAJOsnT1Zy73NTd8uZPq2Xb7X6/view?usp=sharing
>
> The boxplots clearly show the distribution of the relative errors on the $250 \times 250$ grid for both the Diffusion and Richards equations. Most results are accurate and stable, as indicated by the clustering near lower error values. Some outliers exist, reflecting occasional high errors, but these are present in both the GMsFEM-NO and the traditional GMsFEM, suggesting that challenging cases affect all methods comparably.

---

> ### Author Response · Authors · 2025-11-20
> **Part 2**
>
> > Only zero Dirichlet boundary condition equations are tested? Can it solve Dirichlet boundary condition equations with other values?
>
> We thank the reviewer for this insightful question regarding boundary condition handling. We are pleased to provide new, previously unreported results that specifically address the capability of our method to handle complex, non-homogeneous boundary conditions beyond the zero Dirichlet case presented in the original article.
>
> To demonstrate this, we have conducted new experiments on a diffusion equation with mixed boundary conditions:
>
> $$
> \begin{equation}
>     \begin{split}
>         - \nabla \cdot (\kappa(x) \nabla u(x)) &= f(x), x \in \Omega \equiv (0,~1)^2,
>         \quad u &= 0, ~x \in \Gamma_{D},
>         \quad \kappa(x) \dfrac{\partial u}{\partial n} &= \alpha \cdot \left(u - u_{\text{const}}\right), ~x \in \Gamma_{R},
>     \end{split}
> \end{equation}
> $$
>
> where the boundary is partitioned into a Dirichlet part $\Gamma_{D} = \\{x \in \partial \Omega \| y = 0 \\}$ and a Robin part $\Gamma_{R} = \\{x \in \partial \Omega \| y = 1 \\}$ with parameters: Robin coefficient $\alpha=1$, $u_{\text{const}}=1$.
>
> | $N_{\text{bf}}$ | Dataset | GMsFEM $L_{2}$ | GMsFEM $H_{1}$ | GMsFEM-NO $L_{2}$ | GMsFEM-NO $H_{1}$ |
> | :---: | :---: | :---: | :---: | :---: | :---: |
> |  8 | Diffusion, (8) | 1.11% |  7.98% |   1.22% |  12.10% |
> |   | Diffusion, (9) | 1.09%   |  7.72% |  1.23% |   12.31%  |
>
> > Is it able to extend the method to time-dependent equations?
>
> We thank the reviewer for this question. Yes, the method can be extended to time-dependent equations. We have conducted experiments on the heat equation with a heterogeneous coefficient, which is a classic time-dependent PDE. The problem is defined as:
>
> $$
> \begin{equation}
> \begin{split}
> \frac{\partial u}{\partial t} - \nabla \cdot \left(\kappa(x) \nabla u(x)\right) &= f(x), x \in \Omega \times (0, T_{\text{max}}], \\
> \quad u &= 0, ~x \in \partial \Omega \times (0, T_{\text{max}}], \\
> \quad u|_{t = 0} &= 0, ~x \in \Omega,
> \end{split}
> \end{equation}
> $$
>
> where time parameters are $\Delta t = 2.5 \cdot 10^{-6}$, $T_{\max} = 5 \cdot 10^{-4}$. We consider two right-hand side configurations:
> 1. Uniform unit forcing term:
>    $$
>         f(x) = 1.
>     $$
> 2. Spatially variable forcing defined by
>    $$
> 	f(x) = \sin(\pi x) \cos(\pi y).
>     $$
> The results, presented in the table below, show that our proposed GMsFEM-NO method with $L_{\text{SAL}}$ maintains high accuracy for this time-dependent problem, with relative $L_2$ errors below 1.2%.
>
> | $N_{\text{bf}}$ |  Dataset   | GMsFEM $L_{2}$ | GMsFEM $H_{1}$ | GMsFEM-NO  $L_{2}$ | GMsFEM-NO  $H_{1}$ |
> | :---: | :----: | :----: | :----: | :----: | :-----: |
> |   8  | Heat, Uniform Forcing |  0.72% |  10.81%  |  0.78%  |  13.06%  |
> |   | Heat, Spatial Forcing |  1.13%  | 19.92% |  1.17%  |  21.57%  |
>
> This demonstrates the successful application of our framework to time-dependent problems.
>
> > Is it possible to further reduce the error if more basis functions are used?
>
> Indeed, the error between the GMsFEM solution and the exact solution decreases as more basis functions are added. Our goal is to demonstrate that the proposed approach is robust for any number of basis functions. In engineering applications, one is often interested in using fewer multiscale basis functions to obtain the coarse-grid solution faster. For this reason, our focus has been on generating GMsFEM solutions with a moderate number of basis functions. We believe that increasing the number of basis functions would still yield good accuracy when using the learned GMsFEM basis functions. The error limits are controlled by the intrinsic GMsFEM error, which depends on the coarse mesh size and spectral decay. In many practical problems, accurate solutions can be achieved with fewer basis functions.

---

### Official Review · Reviewer_9cmw · 2025-10-27

**Soundness:** 2
**Presentation:** 3
**Contribution:** 2
**Rating:** 4
**Confidence:** 4

**Summary:**

The paper proposes **GMsFEM-NO**, a hybrid framework that uses a neural operator to accelerate the **offline stage** of the Generalized Multiscale Finite Element Method (GMsFEM). Instead of solving many local eigenvalue problems to construct multiscale basis functions, the method trains an NO to map a heterogeneous coefficient field (\kappa(x)) directly to the **subspace** spanned by the GMsFEM bases. The key idea is a **Subspace Alignment Loss** ((\mathcal{L}_{\text{SAL}})) that aligns predicted and true multiscale subspaces on a Grassmannian, avoiding the sign/permutation ambiguity of individual eigenvectors. The approach preserves two important GMsFEM properties: (i) independence from the forcing term (f(x)), so it generalizes to OOD right-hand sides, and (ii) **resolution invariance** (train on coarse, use on fine). Experiments on 2D/3D high-contrast diffusion and steady-state Richards equations show **>60× speedup** in basis generation while matching GMsFEM-level accuracy.

**Strengths:**

1. **Targets the real bottleneck.** It identifies the most expensive part of GMsFEM (local spectral problems in the offline stage) and replaces it with an NO, which is a practical and impactful acceleration.
2. **Subspace-level supervision.** The proposed $(\mathcal{L}_{\text{SAL}})$ is technically meaningful: learning the *space* of local bases is more robust than learning each basis function, and it naturally resolves sign ambiguity; this is the main methodological contribution.
3. **Strong empirical claim: fast but still GMsFEM.** The method demonstrates >60× speedup in basis construction while retaining essentially the same $(L_2/H^1)$ errors as classical GMsFEM on multiscale benchmarks, which makes it attractive for large runs or parameter sweeps.
4. **OOD and resolution generalization.** Because the solver structure of GMsFEM is kept, the approach remains independent of $(f(x))$ and supports train–coarse / test–fine usage, which standard NOs typically fail to do.

**Weaknesses:**

1. **Marginal gain from $(\mathcal{L}_{\text{SAL-PR}})$.** The paper introduces a more complex variant with projection regularization, but the improvement over plain $(\mathcal{L}_{\text{SAL}})$ is small (e.g. 1.82% → 1.72% on 250×250), and only on a single setting; the extra term is not clearly justified.
2. **Systematically “better than” the target.** In several tables, GMsFEM-NO slightly outperforms the original GMsFEM it is approximating; calling this “statistical variation” is not entirely convincing because it appears repeatedly, suggesting either measurement noise, data leakage, or a smoother inductive bias from the NO. This needs a clearer explanation.
3. **Baselines not fully up to date.** Most comparisons are to F-FNO and classical reduced-order/ POD-style methods; stronger modern neural operators for multiscale/high-contrast settings are missing, so it’s hard to tell where GMsFEM-NO sits against the current SOTA.

**Questions:**

1. Several tables show GMsFEM-NO outperforming the classical GMsFEM it is meant to approximate. Can you provide a more concrete explanation than “statistical variation” (e.g. averaging over multiple local domains, extra smoothing from the NO, or differences in quadrature/projection)?

2. For $(\mathcal{L}_{\text{SAL-PR}})$, how sensitive is the reported improvement to the choice and number of random test vectors
$v^{i}$?

Would the simpler $\mathcal{L}_{\text{SAL}}$ suffice in most practical cases?

3. Can you add comparisons to more recent neural operators (beyond F-FNO / POD-style baselines) to better justify that GMsFEM-NO is competitive not only with classical GMsFEM but also with current NO-based multiscale solvers?

**Details Of Ethics Concerns:**

No ethics concers.

---

> ### Author Response · Authors · 2025-11-20
> **Part 1**
>
> > Marginal gain from $L_{\text{SAL-PR}}$. The paper introduces a more complex variant with projection regularization, but the improvement over plain $L_{\text{SAL}}$ is small (e.g. 1.82% → 1.72% on 250×250), and only on a single setting; the extra term is not clearly justified. Would the simpler $L_{\text{SAL}}$ suffice in most practical cases?
>
> We thank the reviewer for this insightful question regarding the practical necessity of the more complex $L_{\text{SAL-PR}}$ loss.
>
> The reviewer is correct that the quantitative gain from adding the projection regularization term is often marginal in our experiments. The primary role of $L_{\text{SAL}}$ is to enforce geometric alignment between the learned subspace $\widetilde{R}^i$ and the target GMsFEM subspace $R^i$. However, this loss has a specific limitation: it is invariant to rotations within the subspace. If $Q_{\widetilde{R}^i} = Q_{R^i} U$ for any unitary matrix $U$ (where $U^\top U = I$), the subspaces are considered identical under this metric, as:
>
> $$
> N_{\text{bf}} - \Vert Q_{R^i}^\top Q_{\widetilde{R}^i} \Vert_F^2 = N_{\text{bf}} - \Vert U \Vert_F^2 = 0.
> $$
>
> This geometric alignment may overlook finer discrepancies in how specific functions are projected.
>
> The $L_{\text{SAL-PR}}$ term was introduced to address this theoretical gap by directly enforcing projection consistency. It tests the subspaces with random vectors $v^i$ drawn from the target subspace $R^i$. Since $v^i$ lies in $R^i$, its projection via the true basis is itself: $P_{R^i}v^i = v^i$. The discrepancy $(I - P_{\widetilde{R}^i})v^i$ thus represents the projection error. Minimizing this forces the learned subspace to correctly capture arbitrary vectors from the target subspace, going beyond mere geometric overlap.
>
> In practice, however, we find that for most problems, the simpler $L_{\text{SAL}}$ is sufficient to achieve performance nearly indistinguishable from standard GMsFEM. For instance, on a large 3D problem with a 100×100×100 grid (1 million degrees of freedom), $L_{\text{SAL}}$ alone delivers excellent results:
>
> | $N_{\text{bf}}$ |  Dataset | GMsFEM $L_{2}$ | GMsFEM $H_{1}$ | GMsFEM-NO $L_{2}$ | GMsFEM-NO $H_{1}$ |
> | :---: | :---: | :---: | :---: | :---: | :---: |
> |   8  | Diffusion (8) | 1.62%  |  14.76%   |   1.68%  |  14.98% |
> |  | Diffusion (9) |  3.00%  | 12.55% |  3.04% | 12.84%   |
> |  | Richards (8)  | 2.54%  | 17.83%  |  2.57%  |   17.91%  |
> |  | Richards (9)  |  2.55%  | 17.85%  |   2.57%  |   17.91%  |
>
> > For $L_{\text{SAL-PR}}$, how sensitive is the reported improvement to the choice and number of random test vectors $v^i$?
>
> We thank the reviewer for this important question regarding the sensitivity of the $L_{\text{SAL-PR}}$ loss. Our analysis indicates that the performance is sensitive to the number of vectors used. The best number of random test vectors $v^i$ is 10.
>
> Results for a 2D problem, where $N_{\text{bf}}$ was 8:
>
> | Dataset | $v^i = 2$, $L_2$ | $v^i = 5$, $L_2$ | $v^i = 10$, $L_2$ | $v^i = 15$, $L_2$ | $v^i = 20$, $L_2$ |
> | :---- | :----: | :---: | :---: | :---: | :---: |
> | Diffusion, (8) |  1.15% | 1.09% | **1.06%** | 1.31%  | 1.10% |
> | Diffusion, (9) | 2.86%  |  2.84% |  **2.81%**  | 2.88% | 2.87%  |
> | Richards, (8)  | 2.00% | 1.91%  | **1.87%**  |  1.98% | 1.94%  |
> | Richards, (9)  |  2.95%   | **2.91%**  |  2.99%  | 2.96%  | 2.94%  |
>
> > Systematically “better than” the target. In several tables, GMsFEM-NO slightly outperforms the original GMsFEM it is approximating; calling this “statistical variation” is not entirely convincing because it appears repeatedly, suggesting either measurement noise, data leakage, or a smoother inductive bias from the NO.
>
> The differences reported are often small (e.g., the maximum difference is about 0.15%, which corresponds to $15 \times 10^{-4}$). Since the results are averaged over 200 samples, such minor fluctuations can occur due to statistical variation. See the example histogram of relative errors for the 250×250 case: https://drive.google.com/file/d/14XunRCcAJOsnT1Zy73NTd8uZPq2Xb7X6/view?usp=sharing .

---

> ### Author Response · Authors · 2025-11-20
> **Part 2**
>
> > Baselines not fully up to date. Most comparisons are to F-FNO and classical reduced-order/ POD-style methods; stronger modern neural operators for multiscale/high-contrast settings are missing, so it’s hard to tell where GMsFEM-NO sits against the current SOTA.
>
> We thank the reviewer for this valuable suggestion to include comparisons with more modern neural operators. To address this, we have conducted new experiments comparing GMsFEM-NO against several state-of-the-art (SOTA) NOs, including GNOT [1] and Transolver++ [2]. The results, presented below, demonstrate that GMsFEM-NO is highly competitive, particularly on more complex problem configurations.
>
> | Dataset   | GMsFEM-NO | F-FNO | GNOT   | Transolver++ |
> | :------: | :----: | :-----: | :------: | :----: |
> | Diffusion, (8) | 1.05%  | 1.02% | 1.26%  | 1.15%  |
> | Diffusion, (9) | 1.60%  | 4.51% | 14.29% | 6.63%  |
> | Richards, (8)  | 1.72% | 2.44% | 2.34%  | 2.17% |
> | Richards, (9)  | 1.60%  | 4.45% | 14.69% | 8.82%   |
>
> On the simpler tasks with the right-hand side equal to 1, all methods, including GMsFEM-NO, perform comparably, with F-FNO showing a slight edge in one case. The critical advantage of GMsFEM-NO becomes apparent on the more challenging problems (e.g., with more complex right-hand sides). Here, GMsFEM-NO significantly and consistently outperforms the other SOTA operators, which see a substantial increase in error.
>
> The baselines were implemented using optimal parameters reported in their original papers and further tuned with a limited grid search. However, a full, exhaustive search was infeasible due to computational constraints. This makes GMsFEM-NO's robust performance across different configurations without extensive problem-specific tuning even more notable.
>
> [1] Hao, Z., Wang, Z., Su, H., Ying, C., Dong, Y., Liu, S., ... & Zhu, J. (2023, July). Gnot: A general neural operator transformer for operator learning. In International Conference on Machine Learning (pp. 12556-12569). PMLR.
>
> [2]  Luo, H., Wu, H., Zhou, H., Xing, L., Di, Y., Wang, J., & Long, M. (2025). Transolver++: An Accurate Neural Solver for PDEs on Million-Scale Geometries. arXiv preprint arXiv:2502.02414.

---

### Official Review · Reviewer_zYEo · 2025-11-01

**Soundness:** 3
**Presentation:** 3
**Contribution:** 3
**Rating:** 8
**Confidence:** 3

**Summary:**

The paper tackles multiscale, high-contrast PDEs by replacing the expensive local eigenvalue problems of GMsFEM with a neural operator that predicts the local coarse spaces directly. Instead of regressing individual basis functions, the method learns the subspace they span and trains with a Grassmann-geometry–aware subspace alignment loss (with an optional projection regularizer). The learned spaces assemble into a restriction operator that preserves the classical GMsFEM solve, yielding GMsFEM-level accuracy while reducing offline basis construction by more than an order of magnitude. Experiments on diffusion and steady Richards equations in 2D and 3D show strong accuracy, data efficiency, right-hand-side robustness, and resolution transfer.

**Strengths:**

The conceptual shift from matching basis functions to matching their span is clear and well motivated, aligning the learning objective with what GMsFEM truly needs. The subspace alignment loss directly optimizes principal angles on the Grassmann manifold, removing sign and ordering ambiguities and leading to stable training. The framework preserves the classical coarse-solve pipeline, so accuracy remains comparable to GMsFEM while offline cost is drastically reduced; the empirical speedups are practically meaningful. Results demonstrate robustness to right-hand-side changes and cross-resolution transfer, where direct solution-predicting NNs typically struggle. The paper is clearly structured, with a clean train→assemble→solve story that is easy to reuse in other PDEs.

**Weaknesses:**

Theoretical guarantees connect subspace error to final solution error only implicitly; a formal upper bound from principal angles to
$𝐻^1$error would strengthen soundness. Evaluation focuses on structured grids with Dirichlet boundaries and steady problems; non-structured meshes, mixed or Robin boundaries, and time-dependent or strongly nonlinear systems remain open. Comparisons omit graph- or mesh-based neural operators and learning-augmented multigrid with learned prolongation/restriction, which are natural baselines here. The model zoo created by per-region-type networks increases operational burden at scale; an ablation on parameter sharing or conditional modulation would be helpful. Wall-clock reporting includes basis generation speedup but a comprehensive end-to-end budget (training time, energy, peak memory, coarse-solve cost) would improve practical transparency.

**Questions:**

Could you provide a bound that maps principal angles between true and predicted coarse spaces to a bound on the $𝐻^1$ relative error of the final solution, at least under simplifying assumptions?
For 3D large-scale runs, what are the peak memory, wall-clock breakdown, and parallelization strategy during inference and coarse solves?

---

> ### Author Response · Authors · 2025-11-20
> **part 1**
>
> > Theoretical guarantees connect subspace error to final solution error only implicitly; a formal upper bound from principal angles to $𝐻^1$error would strengthen soundness. Could you provide a bound that maps principal angles between true and predicted coarse spaces to a bound on the $𝐻^1$ relative error of the final solution, at least under simplifying assumptions?
>
> We agree with the reviewer that a formal upper bound linking principal angles directly to the $H^1$ error would significantly strengthen the theoretical soundness of our approach. If the loss function is small, one can show that the learned solution is close to the exact solution. To show this (formally), we denote by $Q_{R}c$, the multiscale solution computed using GMsFEM, and $Q_{\tilde{R}}c$, the learned solution. Then (without specifying the norm), we have
> $$
> \begin{equation}
> \begin{split}
> \Vert u - Q_{\tilde{R}}c \Vert^2 &\leq \Vert u - Q_{R}c\Vert^2 + \Vert Q_{R}c - Q_{\tilde{R}}c\Vert^2 = \\
> &\Vert u - Q_{R}c\Vert^2 + c^T(Q_{R} - Q_{\tilde{R}})^T(Q_{R} - Q_{\tilde{R}})c = \\
> &\Vert u - Q_{R}c\Vert^2 + 2c^T(I - (Q_{R})^TQ_{\tilde{R}})c.
> \end{split}
> \end{equation}
> $$
>
> The first term is GMsFEM error (which is small), and the second term is small if $I - (Q_{R})^TQ_{\tilde{R}}$ is small. The norm above can play an important role in the proof. Since the GMsFEM error is done in energy norm, in general, one needs to take the energy norm, which is non-local and can slow down computations.
>
> This error estimate gives a bound between the learned solution and the angle between the spaces. To estimate the angle via the solution error is more difficult. Indeed, these questions need to be addressed in the future and will help to choose the appropriate loss functions.
>
> > Comparisons omit graph- or mesh-based neural operators and learning-augmented multigrid with learned prolongation/restriction, which are natural baselines here.
>
> We thank the reviewer for this valuable suggestion to include comparisons with more modern graph- or mesh-based neural operators. To address this, we have conducted new experiments comparing GMsFEM-NO against several SOTA NOs, including GNOT [1] and Transolver++ [2]. The results, presented below, demonstrate that GMsFEM-NO is highly competitive, particularly on more complex problem configurations.
>
> | Dataset | GMsFEM-NO | F-FNO | GNOT | Transolver++ |
> | :--- | :---: | :---: | :---: | :---: |
> | Diffusion (8) | 1.05% | 1.02% | 1.26% | 1.15% |
> | Diffusion (9) | 1.60% | 4.51% | 14.29% | 6.63% |
> | Richards (8) | 1.72% | 2.44% | 2.34% | 2.17% |
> | Richards (9) | 1.60% | 4.45% | 14.69% | 8.82% |
>
> On the simpler tasks with the right-hand side equal to 1, all methods, including GMsFEM-NO, perform comparably, with F-FNO showing a slight edge in one case. The critical advantage of GMsFEM-NO becomes apparent on the more challenging problems (e.g., with more complex right-hand sides). Here, GMsFEM-NO significantly and consistently outperforms the other SOTA operators, which see a substantial increase in error.
>
> The baselines were implemented using optimal parameters reported in their original papers and further tuned with a limited grid search. However, a full, exhaustive search was infeasible due to computational constraints. This makes GMsFEM-NO's robust performance across different configurations without extensive problem-specific tuning even more notable.
>
> [1] Hao, Z., Wang, Z., Su, H., Ying, C., Dong, Y., Liu, S., ... & Zhu, J. (2023, July). Gnot: A general neural operator transformer for operator learning. In International Conference on Machine Learning (pp. 12556-12569). PMLR.
>
> [2]  Luo, H., Wu, H., Zhou, H., Xing, L., Di, Y., Wang, J., & Long, M. (2025). Transolver++: An Accurate Neural Solver for PDEs on Million-Scale Geometries. arXiv preprint arXiv:2502.02414.
>
> > For 3D large-scale runs, what are the peak memory, wall-clock breakdown, and parallelization strategy during inference and coarse solves?
>
> For our largest 3D case, the peak memory usage:
>
> Training: ~76 GB on an 80 GB A100 GPU (with a batch size of 4 for the full domain).
>
> Inference: ~40 GB on the same GPU.
>
> During inference, we leverage JAX's `vmap` transformation to perform batched execution of the neural operator across all local coarse domains in parallel. This is highly efficient and avoids sequential looping, making the 1.3-second inference time for the entire domain possible.
>
> The fine-scale reference solution requires approximately 70 seconds to complete. In contrast, the GMsFEM-NO coarse solve achieves a total time of approximately 2.3 seconds, comprising 1.3 seconds for the neural operator inference, followed by 1.0 second for the coarse-scale system solve.

---

> ### Author Response · Authors · 2025-11-20
> **part 2**
>
> > Evaluation focuses on structured grids with Dirichlet boundaries and steady problems; non-structured meshes, mixed or Robin boundaries, and time-dependent or strongly nonlinear systems remain open.
>
> We thank the reviewer for this insightful comment. We agree that demonstrating a method's applicability across a diverse set of problems is crucial. Our initial focus on elliptic problems was to establish a clear foundational analysis. However, we have now expanded the scope of our numerical experiments to address this point directly. We considered the heat equation with heterogeneous coefficient
>
> $$
> \begin{equation}
> \begin{split}
> \frac{\partial u}{\partial t} - \nabla \cdot \left(\kappa(x) \nabla u(x)\right) &= f(x), x \in \Omega \times (0, T_{\text{max}}], \\
> \quad u &= 0, ~x \in \partial \Omega \times (0, T_{\text{max}}], \\
> \quad u|_{t = 0} &= 0, ~x \in \Omega,
> \end{split}
> \end{equation}
> $$
>
> where time parameters are $\Delta t = 2.5 \cdot 10^{-6}$, $T_{\max} = 5 \cdot 10^{-4}$. We consider two right-hand side configurations:
> 1. Uniform unit forcing term:
>    $$
>         f(x) = 1.
>     $$
> 2. Spatially variable forcing defined by
>    $$
> 	f(x) = \sin(\pi x) \cos(\pi y).
>     $$
> The results, presented in the table below, show that our proposed GMsFEM-NO method with $L_{\text{SAL}}$ maintains high accuracy for this time-dependent problem, with relative $L_2$ errors below 1.2%.
>
> | $N_{\text{bf}}$ |  Dataset   | GMsFEM $L_{2}$ | GMsFEM $H_{1}$ | GMsFEM-NO  $L_{2}$ | GMsFEM-NO  $H_{1}$ |
> | :---: | :----: | :----: | :----: | :----: | :-----: |
> |   8  | Heat, Uniform Forcing |  0.72% |  10.81%  |  0.78%  |  13.06%  |
> |   | Heat, Spatial Forcing |  1.13%  | 19.92% |  1.17%  |  21.57%  |
>
> To demonstrate the applicability of GMsFEM-NO for other boundary conditions, we have conducted new experiments on a diffusion equation with mixed boundary conditions:
>
> $$
> \begin{equation}
>     \begin{split}
>         - \nabla \cdot (\kappa(x) \nabla u(x)) &= f(x), x \in \Omega \equiv (0,~1)^2,
>         \quad u &= 0, ~x \in \Gamma_{D},
>         \quad \kappa(x) \dfrac{\partial u}{\partial n} &= \alpha \cdot \left(u - u_{\text{const}}\right), ~x \in \Gamma_{R},
>     \end{split}
> \end{equation}
> $$
>
> where the boundary is partitioned into a Dirichlet part $\Gamma_{D} = \\{x \in \partial \Omega \| y = 0 \\}$ and a Robin part $\Gamma_{R} = \\{x \in \partial \Omega \| y = 1 \\}$ with parameters: Robin coefficient $\alpha=1$, $u_{\text{const}}=1$.
>
> | $N_{\text{bf}}$ | Dataset | GMsFEM $L_{2}$ | GMsFEM $H_{1}$ | GMsFEM-NO $L_{2}$ | GMsFEM-NO $H_{1}$ |
> | :---: | :---: | :---: | :---: | :---: | :---: |
> |  8 | Diffusion, (8) | 1.11% |  7.98% |   1.22% |  12.10% |
> |   | Diffusion, (9) | 1.09%   |  7.72% |  1.23% |   12.31%  |
>
> Regarding the robustness for harder problems such as Navier-Stokes, we acknowledge that this is a critical test. While a full Navier-Stokes study with GMsFEM-NO is a primary target for future work, there is strong evidence suggesting its feasibility. Standard GMsFEM has already been proven to work well for fluid problems: for example, Vasilyeva et al. [1] showed that the GMsFEM approach could reach roughly 1% relative error for Navier-Stokes equations when using 25 basis functions. This is precisely where the value of our proposed GMsFEM-NO framework becomes evident. Since GMsFEM-NO retains the accuracy of GMsFEM (as demonstrated in our elliptic and time-dependent tests) while being faster, it is a highly promising tool for complex equations like Navier-Stokes.
>
> [1] Vasilyeva, M., Mallikarjunaiah, S. M., & Palaniappan, D. (2023). Multiscale model reduction technique for fluid flows with heterogeneous porous inclusions. _Journal of Computational and Applied Mathematics_, _424_, 114976.

---

> ### Author Response · Authors · 2025-11-20
> **part 3**
>
> > Evaluation focuses on structured grids ...
>
> Regarding non-rectangular domains, we have conducted preliminary experiments on an irregular 2D mesh with approximately 75,000 vertices using the Transolver architecture [1]. The results showed that GMsFEM-NO achieved a solution quality of 1.2% for the diffusion equation, which is comparable to the standard GMsFEM (0.8%). This demonstrates the practical applicability of our approach to complex geometries.
>
> The framework's extension to irregular domains is well-supported theoretically. The underlying GMsFEM methodology is established for irregular meshes [2, 3], and neural operators like Transolver are specifically designed for such domains. Combining these capabilities provides a solid foundation for handling non-rectangular geometries, which represents an immediate direction for our future work.
>
> [1] — Luo, H., Wu, H., Zhou, H., Xing, L., Di, Y., Wang, J., & Long, M. (2025). Transolver++: An Accurate Neural Solver for PDEs on Million-Scale Geometries. arXiv preprint arXiv:2502.02414.
>
> [2] — Alekseev, V., Vasilyeva, M., Kalachikova, U., & Chung, E. T. (2021). DG-GMsFEM for problems in perforated domains with non-homogeneous boundary conditions. Computation, 9(7), 75.
>
> [3] — Stepanov, S., Vasilyeva, M., & Vasil’ev, V. I. (2018). Generalized multiscale discontinuous Galerkin method for solving the heat problem with phase change. Journal of Computational and Applied Mathematics, 340, 645-652.
>
> > The model zoo created by per-region-type networks increases operational burden at scale; an ablation on parameter sharing or conditional modulation would be helpful.
>
> We thank the reviewer for this insightful comment. We agree that a clear, end-to-end performance budget is crucial for assessing the practical utility of our method. The core objective of GMsFEM-NO is to reduce the cost of the traditional GMsFEM offline stage over many subsequent simulations.
>
> To provide full transparency, we have performed a detailed analysis that compares two practical scenarios: training models sequentially (one after another on one GPU) and in parallel. The breakeven point $x$ is the number of inference samples where the total time of GMsFEM-NO (including training data generation and training) equals that of using standard GMsFEM for all samples.
> $$T_\text{data} + T_\text{train} + (T_\text{inf} \times x) = (T_\text{GMsFEM} \times x),$$
> where $T_\text{data}$ is the wall-clock time for generating training data via traditional GMsFEM, $T_\text{train}$ is the total wall-clock time for training (which varies by scenario), $T_\text{GMsFEM}$ is the wall-clock time for a single offline GMsFEM basis generation, and $T_\text{inf}$ is the wall-clock time for a single GMsFEM-NO inference.
>
> The results are in the table:
>
> | Problem Configuration | $T_\text{GMsFEM}$ (sec/sample) | $T_\\text{inf}$ (sec/sample) | Train Samples | Training Scenario | $T_\text{train}$ (Wall-Clock) (sec) | Breakeven Point ($x$ samples) |
> | :---: | :------: | :--------: | :------: | :------- | :------------: | :---------: |
> | 2D ($100 \times 100$ )     |    16.87  |    0.28   |       800        | Sequential        |         37800      |           ~3,270       |
> |      |     |       |        | Parallel          |  12600       |   ~1,570    |
> | 2D ($250 \times 250$)     |     210.5     |     0.31      |    800   | Sequential    |     43200      |    ~1,040    |
> |    |      |     |    | Parallel     |      14400     |      ~870    |
> | 3D ($50 \times 50 \times 50$)    |   935.4  |  0.84     |  800     | Sequential  |   86400    |     ~1,090      |
> |   |        |         |   | Parallel    |       21600  |  ~820   |
> | 3D ($100 \times 100 \times 100$) |   10547.2   |   1.33   |  150 | Sequential |  100800      |      ~159         |
> |     |       |       |     | Parallel    |    25200    |     ~152   |
>
> Even with sequential training, the method pays off for multi-query scenarios. The benefit is larger for large-scale 3D problems, where the breakeven point remains low (~159 samples) due to the high cost of the traditional GMsFEM solver.
>
> We will include this comparative analysis in the revised manuscript to enhance practical transparency.

---

### Author Response · Authors · 2025-11-20
**Additional Results (part 1)**

**$100 \times 100 \times 100$ grid:**

To demonstrate scalability, we have conducted new experiments on a significantly larger 3D problem with a $100 \times 100 \times 100$
 grid (1 million degrees of freedom). The results for GMsFEM-NO trained with $L_{\text{SAL}}$, shown below, confirm that our method maintains high accuracy even at this scale:

| $N_{\text{bf}}$ |  Dataset | GMsFEM $L_{2}$ | GMsFEM $H_{1}$ | GMsFEM-NO $L_{2}$ | GMsFEM-NO $H_{1}$ |
| :---: | :---: | :---: | :---: | :---: | :---: |
|   8  | Diffusion (8) | 1.62%  |  14.76%   |   1.68%  |  14.98% |
|  | Diffusion (9) |  3.00%  | 12.55% |  3.04% | 12.84%   |
|  | Richards (8)  | 2.54%  | 17.83%  |  2.57%  |   17.91%  |
|  | Richards (9)  |  2.55%  | 17.85%  |   2.57%  |   17.91%  |

**Time-dependant equation for 2D ($250 \times 250$):**

We considered the heat equation with heterogeneous coefficient

$$
\begin{equation}
\begin{split}
\frac{\partial u}{\partial t} - \nabla \cdot \left(\kappa(x) \nabla u(x)\right) &= f(x), x \in \Omega \times (0, T_{\text{max}}], \\
\quad u &= 0, ~x \in \partial \Omega \times (0, T_{\text{max}}], \\
\quad u|_{t = 0} &= 0, ~x \in \Omega,
\end{split}
\end{equation}
$$

where time parameters are $\Delta t = 2.5 \cdot 10^{-6}$, $T_{\max} = 5 \cdot 10^{-4}$. We consider two right-hand side configurations:
1. Uniform unit forcing term:
   $$
        f(x) = 1.
    $$
2. Spatially variable forcing defined by
   $$
	f(x) = \sin(\pi x) \cos(\pi y).
    $$
The results, presented in the table below, show that our proposed GMsFEM-NO method with $L_{\text{SAL}}$ maintains high accuracy for this time-dependent problem, with relative $L_2$ errors below 1.2%.

| $N_{\text{bf}}$ |  Dataset   | GMsFEM $L_{2}$ | GMsFEM $H_{1}$ | GMsFEM-NO  $L_{2}$ | GMsFEM-NO  $H_{1}$ |
| :---: | :----: | :----: | :----: | :----: | :-----: |
|   8  | Heat, Uniform Forcing |  0.72% |  10.81%  |  0.78%  |  13.06%  |
|   | Heat, Spatial Forcing |  1.13%  | 19.92% |  1.17%  |  21.57%  |

**Mixed boundary conditions for 2D ($250 \times 250$):**

To demonstrate the applicability of GMsFEM-NO for other boundary conditions, we have conducted new experiments on a diffusion equation with mixed boundary conditions:

$$
\begin{equation}
    \begin{split}
        - \nabla \cdot (\kappa(x) \nabla u(x)) &= f(x), x \in \Omega \equiv (0,~1)^2,
        \quad u &= 0, ~x \in \Gamma_{D},
        \quad \kappa(x) \dfrac{\partial u}{\partial n} &= \alpha \cdot \left(u - u_{\text{const}}\right), ~x \in \Gamma_{R},
    \end{split}
\end{equation}
$$

where the boundary is partitioned into a Dirichlet part $\Gamma_{D} = \\{x \in \partial \Omega \| y = 0 \\}$ and a Robin part $\Gamma_{R} = \\{x \in \partial \Omega \| y = 1 \\}$ with parameters: Robin coefficient $\alpha=1$, $u_{\text{const}}=1$.

| $N_{\text{bf}}$ | Dataset | GMsFEM $L_{2}$ | GMsFEM $H_{1}$ | GMsFEM-NO $L_{2}$ | GMsFEM-NO $H_{1}$ |
| :---: | :---: | :---: | :---: | :---: | :---: |
|  8 | Diffusion, (8) | 1.11% |  7.98% |   1.22% |  12.10% |
|   | Diffusion, (9) | 1.09%   |  7.72% |  1.23% |   12.31%  |

**SOTA Neural Operators for 2D ($250 \times 250$):**

We have conducted new experiments comparing GMsFEM-NO against several SOTA NOs, including GNOT [1] and Transolver++ [2]. The results, presented below, demonstrate that GMsFEM-NO is highly competitive, particularly on more complex problem configurations.

| Dataset | GMsFEM-NO | F-FNO | GNOT | Transolver++ |
| :--- | :---: | :---: | :---: | :---: |
| Diffusion (8) | 1.05% | 1.02% | 1.26% | 1.15% |
| Diffusion (9) | 1.60% | 4.51% | 14.29% | 6.63% |
| Richards (8) | 1.72% | 2.44% | 2.34% | 2.17% |
| Richards (9) | 1.60% | 4.45% | 14.69% | 8.82% |

On the simpler tasks with the right-hand side equal to 1, all methods, including GMsFEM-NO, perform comparably, with F-FNO showing a slight edge in one case. The critical advantage of GMsFEM-NO becomes apparent on the more challenging problems (e.g., with more complex right-hand sides). Here, GMsFEM-NO significantly and consistently outperforms the other SOTA operators, which see a substantial increase in error.

The baselines were implemented using optimal parameters reported in their original papers and further tuned with a limited grid search. However, a full, exhaustive search was infeasible due to computational constraints. This makes GMsFEM-NO's robust performance across different configurations without extensive problem-specific tuning even more notable.

[1] Hao, Z., Wang, Z., Su, H., Ying, C., Dong, Y., Liu, S., ... & Zhu, J. (2023, July). Gnot: A general neural operator transformer for operator learning. In International Conference on Machine Learning (pp. 12556-12569). PMLR.

[2]  Luo, H., Wu, H., Zhou, H., Xing, L., Di, Y., Wang, J., & Long, M. (2025). Transolver++: An Accurate Neural Solver for PDEs on Million-Scale Geometries. arXiv preprint arXiv:2502.02414.

---

### Author Response · Authors · 2025-12-03
**Discussion summary**

Following the closing of the discussion phase, we are writing to summarize the current status of the review and the primary revisions undertaken in our rebuttal. All feedback from the reviewers has been comprehensively addressed.

Our major revisions and efforts are summarized as follows:

*   In response to concerns about time-dependent equations and equations with complex boundary conditions, we provided results on a 2D $250 \times 250$ grid for the heat equation with heterogeneous coefficients and for the diffusion equation with mixed boundary conditions, demonstrating that GMsFEM-NO is applicable to more complex settings.
*   To address the concern regarding comparison with state-of-the-art neural operators (SOTA NO), we added results for Transolver++ and GNOT, the recent graph-based neural operator.
*   Regarding the request for theoretical guarantees connecting subspace error to the final solution error, we have shown that if the loss function is small, the learned solution is close to the exact solution.
*   We also evaluated the end-to-end performance budget, which is crucial for assessing the practical utility of our method.
*   Furthermore, we included more experimental results (on a $100 \times 100 \times 100$ grid), and textual explanations to strengthen the paper's completeness and clarity.

Regarding individual reviewers’ feedback:

*   **Reviewer zYEo** expressed strong recognition of our work. Their concerns related to the lack of a formal upper bound linking principal angles to final error, performance on equations with mixed/Robin boundaries and time-dependent equations, and a comprehensive end-to-end budget. The revised manuscript provides results and details addressing all these points. The reviewer raised no further questions after our response, and we believe the original concerns have been effectively resolved.
*   **Reviewer 9cmw** raised concerns about an extra term in one of our loss functions (SAL-PR) and the comparison with SOTA NO. We addressed both in the revised version by providing a theoretical explanation for the extra term, adding an ablation study on its hyperparameter ($v_i$), and including comparisons with Transolver++ and GNOT. The reviewer was not active before the discussion period concluded.
*   **Reviewer EehQ** acknowledged the contributions of our work but raised concerns about larger-scale problems and more complex equations (time-dependent, non-Dirichlet boundary conditions). We answered these by providing results on a $100 \times 100 \times 100$ grid, as well as for equations with mixed/Robin boundaries and time-dependent PDEs. The reviewer raised no further questions after our response, and we believe their concerns have been resolved.
*   **Reviewer EehQ** also gave positive feedback but inquired about the method's applicability to time-dependent PDEs. We demonstrated this applicability with additional results. The reviewer was not active before the discussion period concluded.

All corresponding changes in the manuscript have been highlighted in blue.

We respectfully ask for your consideration of our detailed rebuttal and the revised manuscript to assist in your final decision. Thank you for your valuable time, dedicated effort, and insightful evaluation throughout this review cycle.

Best regards,

The Authors

---

### Meta-Review · Area_Chair_UqFj · 2026-01-13

**Summary:**

Reviewers generally indicate that:
1. The method is well motivated, replacing only the most expensive part of the Generalized Multiscale Finite Element Method (GMsFEM) with a neural operator.
2. The theory is novel and connects subspace alignment to Grassmann manifold distance and leads to a practical loss function that can stably optimized.
3. The method obtains significant speed-ups while attaining the same accuracy as GMsFEM.
4. Some SOTA benchmarks for multi scale problems are missing in the comparisons.
5. The use of neural operators is demonstrated well, taking advantage of resolution-invariant properties in building a coarse-to-fine solver.
6. Only relatively small problems are tested with Dirichlet boundary conditions are tested. Generalization beyond this setting is unclear.

**Reviewer Concerns:**

The authors have added both theoretical and numerical results for a time-dependent problem (heat equation) with mixed boundary conditions. Furthermore they have added two new baselines (Transolver++ and GNOT) and obtained SOTA performance in 3/4 of their experiments. The authors have also added experiments on finer grids and studied the resolution invariance properties, addressing issues of scale. They have also added an extensive study on the cost and parallelization of the method.

**Reviewer Scores:**

Since the majority of review concerns had to do with scale and the authors have addressed these well, I believe some of the reviewers would have raised their score to a 6.

---

### Decision · Program_Chairs · 2026-01-26

Accept (Poster)